# Lysosomal Ca$^{2+}$-mediated TFEB activation modulates mitophagy and functional adaptation of pancreatic β-cells to metabolic stress

Kihyoun Park [1,2], Hyejin Lim [1], Jinyoung Kim [1], Yeseong Hwang[1], Yu Seol Lee[1], Soo Han Bae [1], Hyeongseok Kim[3], Hail Kim[4], Shin-Wook Kang[5], Joo Young Kim [6] & Myung-Shik Lee [1,5,7 ✉]

Although autophagy is critical for pancreatic β-cell function, the role and mechanism of mitophagy in β-cells are unclear. We studied the role of lysosomal Ca$^{2+}$ in TFEB activation by mitochondrial or metabolic stress and that of TFEB-mediated mitophagy in β-cell function. Mitochondrial or metabolic stress induced mitophagy through lysosomal Ca$^{2+}$ release, increased cytosolic Ca$^{2+}$ and TFEB activation. Lysosomal Ca$^{2+}$ replenishment by ER‑>lysosome Ca$^{2+}$ refilling was essential for mitophagy. β-cell-specific *Tfeb* knockout (*Tfeb*$^{Δβ-cell}$) abrogated high-fat diet (HFD)-induced mitophagy, accompanied by increased ROS and reduced mitochondrial cytochrome *c* oxidase activity or O$_2$ consumption. *Tfeb*$^{Δβ-cell}$ mice showed aggravation of HFD-induced glucose intolerance and impaired insulin release. Metabolic or mitochondrial stress induced TFEB-dependent expression of mitophagy receptors including *Ndp52* and *Optn*, contributing to the increased mitophagy. These results suggest crucial roles of lysosomal Ca$^{2+}$ release coupled with ER‑>lysosome Ca$^{2+}$ refilling and TFEB activation in mitophagy and maintenance of pancreatic β-cell function during metabolic stress.

[1] Severance Biomedical Science Institute, Yonsei University College of Medicine, Seoul, Korea. [2] Department of Health Sciences and Technology, SAIHST, Sungkyunkwan University School of Medicine, Seoul, Korea. [3] Department of Biochemistry, Chungnam National University, Daejeon, Korea. [4] Graduate School of Medical Science and Engineering, Korea Advanced Institute of Science and Technology, Daejeon, Korea. [5] Department of Internal Medicine, Yonsei University College of Medicine, Seoul, Korea. [6] Department of Pharmacology and Brain Korea 21 Project for Medical Sciences, Seoul, Korea. [7] Soonchunhyang Institute of Medi-bio Science, Soonchunhyang University, Cheonan, Korea. ✉email: mslee0923@yuhs.ac

Mitochondrial function is essential for insulin release and survival of pancreatic β-cells[1]. Mitochondrial function is maintained by several factors such as mitochondrial biogenesis, fission, fusion, and selective autophagy (mitophagy). The importance of autophagy in the maintenance of β-cell function and mitochondria has been demonstrated in previous studies using β-cell-specific autophagy knockout (KO) ($Atg7^{\Delta\beta\text{-cell}}$) mice which had reduced β-cell mass and mitochondrial abnormalities[2]. However, the role and mechanism of mitophagy in pancreatic β-cells have not been clearly demonstrated.

Transcription factor EB (TFEB) acts as a master regulator of lysosome biogenesis and autophagy gene expression[3]. TFEB family members also play a role in mitophagy[4,5]. In TFEB activation and mitophagy by mitochondrial stressors, lysosomal $Ca^{2+}$ release plays an important part[5]. However, the mechanism that perpetuates lysosomal $Ca^{2+}$ release and enables the full progression of mitophagy despite the small lysosomal volume[6] is not clear.

We investigated the mechanism and functional role of mitophagy in pancreatic β-cells and found that TFEB is activated by mitochondrial reactive oxygen species (ROS)-induced lysosomal $Ca^{2+}$ release through calcineurin. We also obtained data indicating the importance of ER → lysosome $Ca^{2+}$ refilling in the sustained release of lysosomal $Ca^{2+}$, TFEB activation and mitophagy by mitochondrial stressors. Finally, we observed that TFEB contributes to β-cell mitophagy through induction of mitophagy receptors such as *Calcoco2* (*Ndp52*) and *Optn*.

## Results

### Mitophagy and TFEB nuclear translocation by mitochondrial stressors.

We first studied whether mitophagy occurs in INS-1 insulinoma cells treated with mitochondrial stressors. Immunoblotting showed that INS-1 cells express Parkin (Supplementary Fig. 1a), which is necessary for mitophagy by mitochondrial stressors[4]. When we transfected cells with *pMito-Keima* encoding a pH-sensitive fluorescent probe with a mitochondria-targeting sequence[7], tubular mitochondrial Keima fluorescence excited by 440-nm laser was observed. After rotenone treatment, a mitochondrial complex I inhibitor, many cells showed punctate fluorescence excited by a 590-nm laser (acidic puncta), indicating lysosomal delivery of mitochondrial Keima and mitophagy (Fig. 1a). When cells were treated with another mitochondrial stressor antimycin A, a mitochondrial complex III inhibitor, combined with oligomycin, an ATP synthase inhibitor (O/A)[4], acidic Mito-Keima puncta were again well observed, indicating mitophagy (Fig. 1a). To confirm mitophagy, cells were transfected with *mRFP-LC3* and treated with rotenone or O/A. In many cells, mRFP-LC3 puncta colocalized with TOM20, a mitochondrial outer membrane protein, were observed by immunofluorescence (Fig. 1b), supporting rotenone- or O/A-induced mitophagy in INS-1 cells.

We next studied the role of TFEB, a master regulator of lysosomal biogenesis and autophagy genes[3], in mitophagy because transcriptional control of autophagy/lysosomal genes could be important in mitophagy[4]. When we studied TFEB nuclear translocation, a crucial event in TFEB activation, using *TFEB-GFP*-transfected INS-1 cells, TFEB nuclear fluorescence was apparent after rotenone or O/A treatment (Fig. 1c), similar to previous results using HeLa cells[4]. When INS-1 cells transfected with *GFP* conjugated to *TFE3*, another member of the MiTF/TFE family (*TFE3-GFP*)[8], TFE3 nuclear fluorescence was clearly observed after rotenone or O/A treatment as well (Fig. 1c). Nuclear translocation of endogenous TFEB and TFE3 was also observed after rotenone or O/A treatment (Fig. 1d). To study the role of TFEB/TFE3 activation in mitophagy, *Tfeb*- or *Tfe3*-KO

INS-1 cells generated by CRISP3/Cas9 technology were employed[9] (Supplementary Fig. 1b). In these cells, mitophagy by rotenone or O/A was significantly reduced (Fig. 1e), showing an important role of *Tfeb* or *Tfe3* in mitophagy by mitochondrial stressors.

### Lysosomal $Ca^{2+}$ in mitophagy.

To study the mechanism of TFEB nuclear translocation by mitochondrial stressors, we examined whether mitochondrial stressor-induced mitophagy involves calcineurin, one of the most important phosphatases regulating TFEB activation in starvation[10]. When INS-1 cells transfected with *TFEB-GFP* and *HA-ΔCnA-H151Q*, a dominant-negative mutant of calcineurin[11], were treated with rotenone or O/A, TFEB nuclear translocation by mitochondrial stressors was significantly reduced (Fig. 2a, b), indicating the role of calcineurin in TFEB activation by mitochondrial stressors. We also studied the role of calcineurin in mitophagy. In cells transfected with *mRFP-LC3* and *HA-ΔCnA-H151Q* and then treated with rotenone or O/A, mitophagy identified as mRFP-LC3 puncta colocalized with TOM20, was also significantly reduced (Fig. 2c, d), showing the role of calcineurin in mitophagy by mitochondrial stressors.

We next studied whether lysosomal $Ca^{2+}$ release is involved in TFEB activation by mitochondrial stressors[10]. When INS-1 cells transfected with *GCaMP3-ML1* encoding a lysosome-specific $Ca^{2+}$ probe detecting perilysosomal $Ca^{2+}$[12] were treated with rotenone or O/A, lysosomal $Ca^{2+}$ release was not visualized; however, perilysosomal $Ca^{2+}$ release by Gly-Phe β-naphthylamide, (GPN), a cathepsin C substrate inducing osmotic lysis of lysosomes[12], was significantly reduced by rotenone or O/A pretreatment, suggesting preemptying or release of lysosomal $Ca^{2+}$ by rotenone or O/A (Fig. 2e), similar to results using carbonyl cyanide m-chlorophenylhydrazone (CCCP), a mitochondrial uncoupler[5]. We next studied changes in lysosomal $Ca^{2+}$ content ($[Ca^{2+}]_{Lys}$) using Oregon Green 488 BAPTA-1 Dextran. $[Ca^{2+}]_{Lys}$ of INS-1 cells was significantly reduced by rotenone or O/A (Fig. 2f), indicating mitochondrial stressor-induced lysosomal $Ca^{2+}$ release. Indirect estimation of $[Ca^{2+}]_{Lys}$ by calculating area under the curve (AUC) of cytoplasmic $Ca^{2+}$ fluorescence ($[Ca^{2+}]_i$) tracing using Fura-2 after GPN treatment[13], also showed significantly reduced $[Ca^{2+}]_{Lys}$ by rotenone or O/A (Fig. 2g). However, lysosomal pH was not changed by rotenone treatment, while O/A treatment induced a small increase of lysosomal pH (Supplementary Fig. 2a), suggesting that $[Ca^{2+}]_{Lys}$ decrease by mitochondrial stressor is unrelated to lysosomal pH changes. We next investigated whether rotenone or O/A increases $[Ca^{2+}]_i$ due to lysosomal $Ca^{2+}$ release. $[Ca^{2+}]_i$ determined by a ratiometric method using Fura-2 was significantly increased after rotenone or O/A treatment (Fig. 2h), suggesting that lysosomal $Ca^{2+}$ release by rotenone or O/A leads to increased $[Ca^{2+}]_i$, calcineurin activation and TFEB activation. To confirm the role of increased $[Ca^{2+}]_i$ in TFEB translocation and mitophagy, we used BAPTA-AM, a cell-permeable $Ca^{2+}$ chelator[14]. BAPTA-AM pretreatment significantly reduced the proportion of cells with nuclear TFEB and the number of acidic Mito-Keima puncta by rotenone or O/A (Fig. 2i and Supplementary Fig. 3a), indicating TFEB nuclear translocation and mitophagy by cytoplasmic $Ca^{2+}$ released from lysosome.

Since increased $[Ca^{2+}]_i$ by rotenone or O/A induced TFEB nuclear translocation and mitophagy through calcineurin, a phosphatase, we next studied changes in TFEB dephosphorylation by rotenone or O/A. After treatment with rotenone or O/A, electrophoretic mobility of total TFEB was markedly increased, suggesting TFEB dephosphorylation (Supplementary Fig. 4a). Consistently, phosphorylation at S142, one of the most important

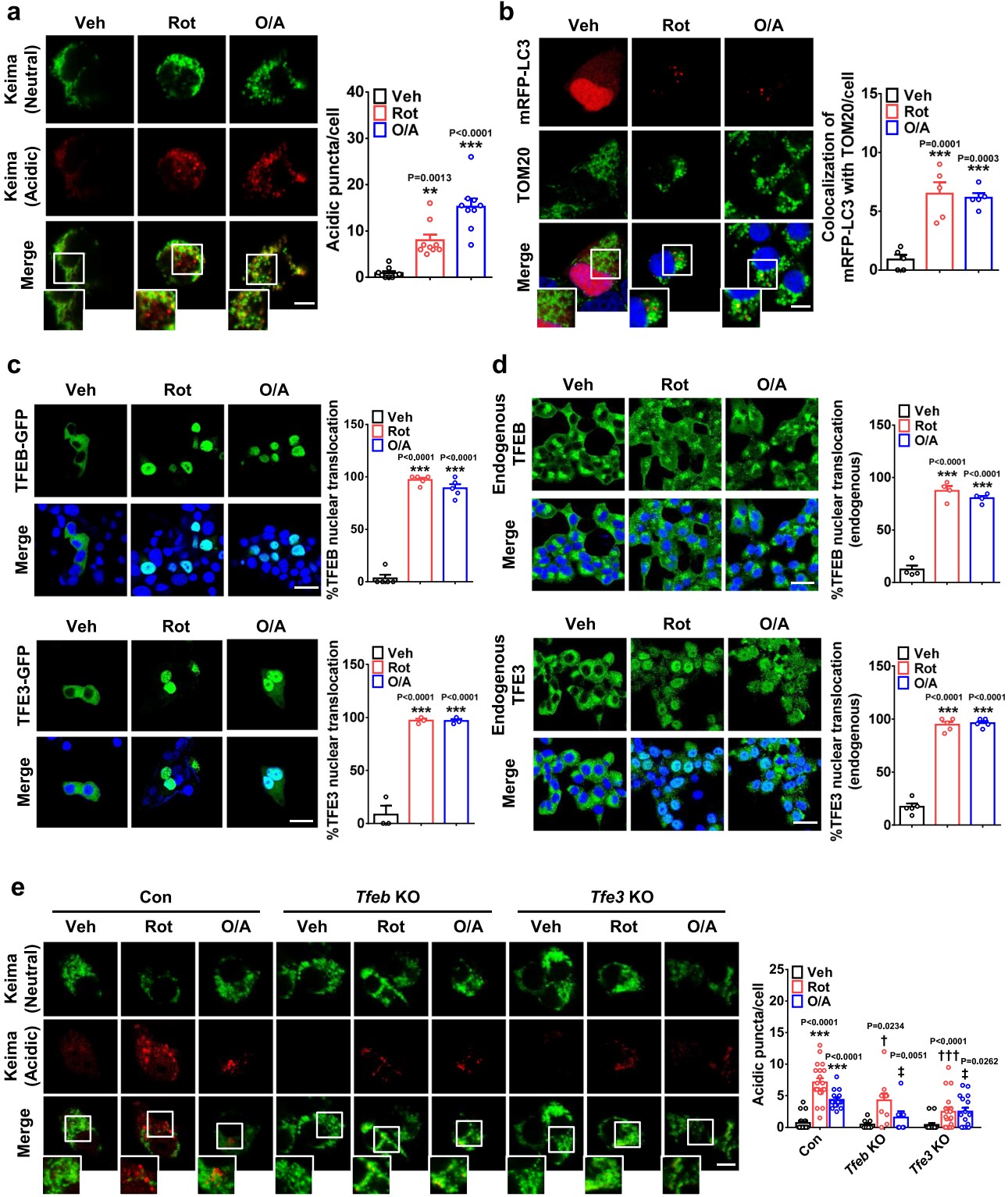

TFEB phosphorylation sites[3], was markedly reduced by rotenone or O/A (Supplementary Fig. 4a). We also studied phosphorylation at S211 of TFEB, another important phosphorylation site, using immunoprecipation assay based on TFEB pS211 binding to 14-3-3 protein[15]. In immunoblotting, band intensity of 14-3-3 binding motif of TFEB, thus pS211-TFEB level[15], in TFEB immunoprecipate was markedly reduced by rotenone or O/A treatment of *TFEB-GFP*-transfectants (Supplementary Fig. 4b). Similarly, band intensity of 14-3-3 binding motif of TFE3, thus

pS321-TFE3 level[16], in TFE3 immunoprecipitate was markedly reduced by rotenone or O/A treatment of *TFE3-GFP*-transfectants (Supplementary Fig. 4b). In addition, we reproducibly observed reduced expression of total TFEB or TFE3 after O/A treatment (Supplementary Fig. 4b), suggesting possible degradation of TFEB or TFE3, while the mechanism and significance of these results are unknown.

We next studied the role of TRPML1 channel, a lysosomal $Ca^{2+}$ channel that can be crucial in autophagy[17]. ML-SI3, a

**Fig. 1 Mitophagy after treating INS-1 cells with mitochondrial stressors. a** INS-1 cells transfected with *pMito-Keima* were treated with rotenone or O/A for 18 h. Keima fluorescence at neutral pH (green) and in acidic conditions (red) was determined, and the number of red puncta indicating mitophagy was counted (right). Representative fluorescence images are presented (left). (scale bar, 5 μm) ($n = 9$) (Veh, vehicle; Rot, rotenone; O/A, oligomycin + antimycin A) **b** INS-1 cells transfected with *mRFP-LC3* were treated with rotenone or O/A for 24 h and then subjected to immunofluorescence using anti-TOM20 Ab to visualize the colocalization of an autophagic marker with a mitochondrial protein, i.e., mitophagy (right). Representative fluorescence images are presented (left). (scale bar, 5 μm) ($n = 5$) **c** INS-1 cells transfected with *TFEB-GFP* or *TFE3-GFP* were treated with rotenone or O/A for 4 h, and then cells with nuclear translocation of TFEB-GFP or TFE3-GFP were counted by confocal microscopy (right). Representative fluorescence images are presented (left). (scale bar, 20 μm) ($n = 5$ for TFEB-GFP; $n = 3$ for TFE3-GFP) **d** INS-1 cells were treated with rotenone or O/A for 4 h, and then cells with total or partial nuclear translocation of endogenous TFEB or TFE3 were counted after immunostaining with anti-TFEB or -TFE3 Ab (right). Representative fluorescence images are presented (left). (scale bar, 20 μm) ($n = 4$ for TFEB; $n = 5$ for TFE3) **e** *Tfeb*-KO and *Tfe3*-KO INS-1 cells generated by CRISPR/Cas9 technology were transfected with *pMito-Keima*, and then treated with rotenone or O/A for 18 h. Keima fluorescence at neutral pH and in acidic condition was determined by confocal microscopy (left). The number of red puncta indicating mitophagy was counted (right). (scale bar, 5 μm) ($n = 7$ for *Tfeb* KO + O/A; $n = 9$ for *Tfeb* KO; $n = 10$ for *Tfeb* KO + Rot; $n = 15$ for O/A or *Tfe3* KO; $n = 16$ for *Tfe3* KO + O/A; $n = 18$ for *Tfe3* KO + Rot; $n = 20$ for Con or Rot) (Con, Cas9 control) Cells in the rectangles were magnified. All data in this figure are the means ± SEM from more than 3 independent experiments. *P* values were determined using one-way ANOVA with Tukey's test. *, compared to Veh-treated cells; †, compared to Rot-treated Con cells; ‡, compared to O/A-treated Con cells.

TRPML1 channel antagonist[18], significantly reduced $[Ca^{2+}]_i$ increase by rotenone or O/A (Supplementary Fig. 3b), suggesting $Ca^{2+}$ release by mitochondrial stressors through TRPML1. ML-SI3 also significantly reduced mitophagy by rotenone or O/A (Supplementary Fig. 3a). To obtain genetic evidence, we conducted knockdown (KD) of *Mcoln1* encoding TRPML1. RT-PCR confirmed markedly reduced *Mcoln1* expression by si*Mcoln1* transfection (Supplementary Fig. 3c). In *Mcoln1*-KD cells, rotenone- or O/A-induced mitophagy detected as acidic Mito-Keima puncta was significantly reduced (Supplementary Fig. 3d), demonstrating a crucial role of TRPML1 in mitochondrial stressor-induced mitophagy.

We next studied the mechanism of lysosomal $Ca^{2+}$ release through TRPML1 by mitochondrial stressors. We examined ROS accumulation because ROS affects diverse ion channels, and can be generated by mitochondrial stressors in the vicinity of lysosome[19]. We observed strong ROS accumulation stained with CM-H2DCFDA by rotenone or O/A (Supplementary Fig. 5a). Pretreatment with *N*-acetylcysteine (NAC), an antioxidant, markedly reduced ROS accumulation by rotenone or O/A (Supplementary Fig. 5a). Intriguingly, NAC significantly inhibited rotenone- or O/A-mediated increase of $[Ca^{2+}]_i$ (Supplementary Fig. 5b), which suggests that ROS induces lysosomal $Ca^{2+}$ release and increases $[Ca^{2+}]_i$, leading to TFEB nuclear translocation. We also studied changes in mitochondrial ROS that can be more relevant to the induction of mitophagy, using MitoSOX. Again, mitochondrial ROS was increased by rotenone or O/A, which was significantly reduced by MitoTEMPO, a mitochondrial ROS scavenger (Supplementary Fig. 5c). Furthermore, both the increase of $[Ca^{2+}]_i$ and the decrease of $[Ca^{2+}]_{Lys}$ by rotenone or O/A were suppressed by MitoTEMPO (Supplementary Fig. 5d, e), indicating the role of mitochondrial ROS in lysosomal $Ca^{2+}$ efflux, increased $[Ca^{2+}]_i$ and decreased $[Ca^{2+}]_{Lys}$ by mitochondrial stressors.

**Role of ER → lysosome Ca²⁺ refilling in mitophagy.** Although $[Ca^{2+}]_{Lys}$ is comparable to $[Ca^{2+}]_{ER}$, lysosome is a minor $Ca^{2+}$ reservoir because of a small lysosomal volume and might not be a sufficient source to ensure full progression of intracellular events requiring $Ca^{2+}$[6]. Therefore, we studied whether ER → lysosome $Ca^{2+}$ movement that can follow lysosomal $Ca^{2+}$ depletion[20], occurs during mitophagy. We observed that decreased $[Ca^{2+}]_{Lys}$ by rotenone or O/A recovered after mitochondrial stressor removal (Fig. 3a). Since ER, the largest $Ca^{2+}$ reservoir in cells, is the most likely source replenishing depleted lysosomal $Ca^{2+}$ store[6], we conducted blockade of ER $Ca^{2+}$ exit channels that could be routes for ER → lysosome $Ca^{2+}$ movement. Xesto-

spongin C, an inositol trisphosphate receptor (InsP3R) antagonist, significantly suppressed $[Ca^{2+}]_{Lys}$ recovery after mitochondrial stressor removal (Fig. 3a), suggesting the occurrence of ER → lysosome $Ca^{2+}$ refilling during mitophagy through InsP3R. Dantrolene, an antagonist of ryanodine receptor, another ER $Ca^{2+}$ exit channel, similarly inhibited $[Ca^{2+}]_{Lys}$ recovery after mitochondrial stressor removal (Fig. 3a), suggesting the involvement of both InsP3R and ryanodine receptor channels in ER → lysosome $Ca^{2+}$ refilling in mitophagy. We next studied the effect of ER $Ca^{2+}$ exit channel blockade on $[Ca^{2+}]_i$ increase by mitochondrial stressors. Both Xesto-spongin C and Dantrolene significantly attenuated $[Ca^{2+}]_i$ increases by mitochondrial stressors (Fig. 3b), suggesting that ER → lysosome $Ca^{2+}$ refilling contributes to the increase of $[Ca^{2+}]_i$ by mitochondrial stressors. When we chelated ER $Ca^{2+}$ with a membrane-permeant chelator N,N,N′,N′-tetrakis (2-pyridylmethyl)ethylene diamine (TPEN) that cannot chelate cytosolic $Ca^{2+}$ due to low $Ca^{2+}$ affinity[21], both $[Ca^{2+}]_{Lys}$ recovery after mitochondrial stressor removal and $[Ca^{2+}]_i$ increase by mitochondrial stressors were reduced (Fig. 3a, b), supporting the role of ER $Ca^{2+}$ in these processes. Furthermore, the number of Mito-Keima puncta after rotenone or O/A treatment was significantly reduced by Xesto-spongin C or Dantrolene (Fig. 3c), probably due to insufficient $Ca^{2+}$ refilling. However, these results do not confirm the role of ER $Ca^{2+}$ in these processes because Xesto-spongin C or Dantrolene might have off-target effects and TPEN can chelate $Zn^{2+}$ in addition to ER $Ca^{2+}$[22]. To further corroborate the role of ER $Ca^{2+}$ in $[Ca^{2+}]_{Lys}$ recovery after mitochondrial stressor removal, we treated cells with Thapsigargin emptying ER $Ca^{2+}$ store[20]. After Thapsigargin pretreatment, $[Ca^{2+}]_{Lys}$ recovery after mitochondrial stressor removal was abrogated (Supplementary Fig. 6a), supporting the role of ER $Ca^{2+}$ in $[Ca^{2+}]_{Lys}$ recovery and ER → lysosome $Ca^{2+}$ refilling.

Since these results suggest ER → lysosome $Ca^{2+}$ movement during mitochondrial stress, we next measured ER $Ca^{2+}$ content ($[Ca^{2+}]_{ER}$) during mitophagy in INS-1 cells transfected with *GEM-CEPIA1er*[23]. When $[Ca^{2+}]_{ER}$ was determined after rotenone or O/A treatment in a $Ca^{2+}$-free Krebs-Ringer bicarbonate (KRB) buffer inhibiting possible store-operated $Ca^{2+}$ entry (SOCE), a significant decrease of $[Ca^{2+}]_{ER}$ was observed (Supplementary Fig. 6b), consistent with ER → lysosome $Ca^{2+}$ refilling. $[Ca^{2+}]_{ER}$ determined using a FRET-based *D1ER* probe allowing ratiometric measurement[24] also demonstrated a significantly reduced $[Ca^{2+}]_{ER}$ after rotenone or O/A treatment in a $Ca^{2+}$-free KRB buffer (Fig. 3d). Importantly, $[Ca^{2+}]_{ER}$ was not changed by rotenone or O/A treatment in the presence of extracellular $Ca^{2+}$ (Supplementary Fig. 6c). These results

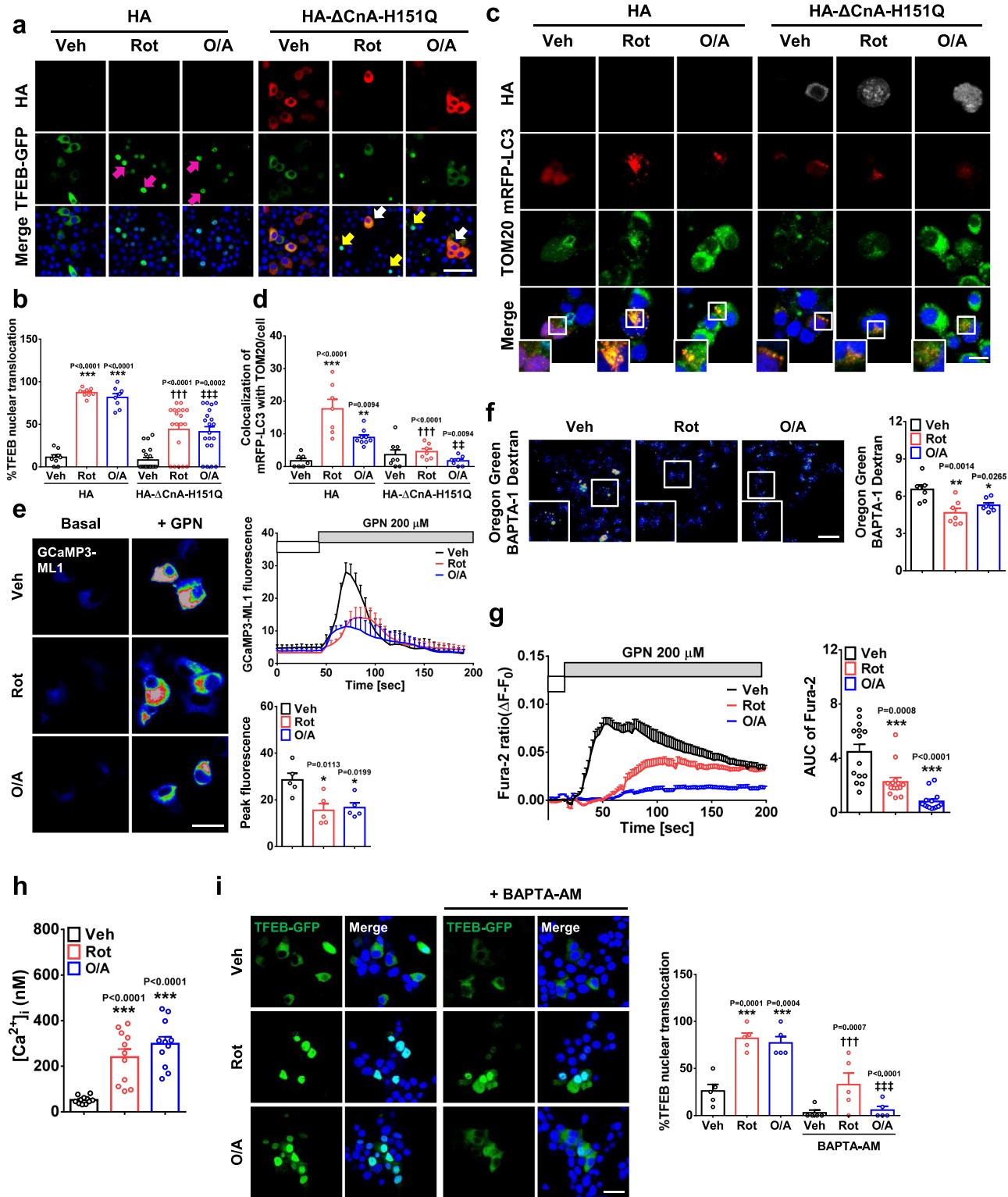

eliminate possible role of sarcoplasmic/endoplasmic reticulum $Ca^{2+}$-ATPase (SERCA) inhibition due to rotenone- or O/A-induced decrease of ATP in $[Ca^{2+}]_{ER}$ decrease after mitochondrial stressor treatment without extracellular $Ca^{2+}$ (see Fig. 3d), since $[Ca^{2+}]_{ER}$ would be decreased by SERCA inhibition even with extracellular $Ca^{2+}$[23].

To study dynamic changes in $[Ca^{2+}]_{ER}$ and their temporal relationship with lysosomal $Ca^{2+}$ refilling, we simultaneously traced $[Ca^{2+}]_{ER}$ and $[Ca^{2+}]_{Lys}$ in cells transfected with *GEM-*

*CEPIA1er* and loaded with Oregon Green 488 BAPTA-1 Dextran. When cells were treated with rotenone or O/A and organelle $[Ca^{2+}]$ of cells with reduced $[Ca^{2+}]_{Lys}$ was monitored without rotenone or O/A in a $Ca^{2+}$-free KRB buffer, $[Ca^{2+}]_{ER}$ decrease occurred simultaneously with $[Ca^{2+}]_{Lys}$ recovery (Fig. 3e), strongly supporting ER → lysosome $Ca^{2+}$ refilling by mitochondrial stress. Without mitochondrial stressor removal, $[Ca^{2+}]_{Lys}$ recovery could not be observed probably due to ongoing lysosomal $Ca^{2+}$ release (Supplementary Fig. 6d).

**Fig. 2 Role of Ca$^{2+}$ and calcineurin in mitophagy. a** *Tfeb-GFP*-transfectants were transfected with *pcDNA 3.1-HA* or *-HA-ΔCnA-H151Q*. After rotenone or O/A treatment for 4 h, cells were subjected to immunofluorescence with anti-HA Ab. (scale bar, 20 μm) (white arrow, *HA-ΔCnA-H151Q*-transfected cells showing no TFEB translocation by rotenone or O/A; yellow arrow, untransfected cells showing TFEB translocation by rotenone or O/A; magenta arrow, control-transfected cells showing TFEB translocation by rotenone or O/A). **b** The percentage of cells showing nuclear TFEB in cells of (**a**). (scale bar, 20 μm) ($n = 8$ for HA + Rot or HA + O/A; $n = 9$ for HA + Veh; $n = 19$ for HA-ΔCnA-H151Q + Rot; $n = 20$ for HA-ΔCnA-H151Q + O/A; $n = 21$ for HA-ΔCnA-H151Q + Veh) **c** *mRFP-LC3*-transfectants were transfected with *pcDNA 3.1-HA* or *-HA-ΔCnA-H151Q*. After rotenone or O/A treatment for 24 h, cells were subjected to immunofluorescence with anti-TOM20 and -HA Abs. (scale bar, 10 μm) **d** The number mRFP-LC3 puncta colocalized with TOM20 in cells of (**c**). ($n = 7$ for HA + Veh, HA + Rot, HA-ΔCnA-H151Q + Rot or HA-ΔCnA-H151Q + O/A; $n = 8$ for HA-ΔCnA-H151Q + Veh; $n = 9$ for HA + O/A) **e** During GPN treatment of *GCaMP3-ML1*-transfected cells with or without rotenone or O/A pretreatment, perilysosomal fluorescence was traced (right upper). Peak fluorescence (right lower). Representative fluorescence images (left). (scale bar, 20 μm) ($n = 5$) **f** After mitochondrial stressor treatment for 4 h, [Ca$^{2+}$]$_{Lys}$ was determined by Oregon Green 488 BAPTA-1 Dextran loading. Fluorescence intensity (right). Representative fluorescence images (left). (scale bar, 20 μm) ($n = 7$) **g** During GPN treatment of Fura-2-loaded cells with or without rotenone or O/A pretreatment, 340/380 nm fluorescence ratio was monitored (left). Lysosomal Ca$^{2+}$ reservoir estimated by calculating AUC of the curve (right). ($n = 14$) **h** After treatment with mitochondrial stressors for 1 h, [Ca$^{2+}$]$_i$ was determined using Fura-2. ($n = 10$) **i** After mitochondrial stressor treatment for 4 h in the presence or absence of BAPTA-AM, TFEB-GFP nuclear translocation was determined (right). Representative fluorescence images (left). (scale bar, 20 μm) ($n = 5$) Cells in the rectangles were magnified. All data in this figure are the means ± SEM from more than 3 independent experiments. *P* values were determined using one-way ANOVA with Tukey's test. *, compared to Veh-treated cells; †, compared to INS-1 cells or HA transfectants treated with Rot alone; ‡, compared to INS-1 cells or HA transfectants treated with O/A alone.

Since ER → lysosome Ca$^{2+}$ refilling can be facilitated by membrane contact between organelles[25], we studied possible apposition of ER and lysosome proteins using proximity ligation assay (PLA). Confocal microscopy after incubation with specific antibodies (Abs) and rolling circle amplification showed close contact between VAPA on ER and ORP1L on lysosome[26] by rotenone or O/A (Fig. 3f), suggesting the formation of membrane contact sites between ER and lysosome by mitochondrial stressors, likely facilitating ER → lysosome Ca$^{2+}$ refilling. We employed another method detecting ER-lysosome contact-a modification of Contact-ID technology using a split–pair system of pBir biotin ligase[27]. When cells transfected with Flag-BirA(N-G78)-VAPA and STARD3-BirA(G79-C)-HA were treated with rotenone or O/A, biotin conjugate formation was observed after incubation with FITC-streptavidin, indicating contact between ER (VAPA) and lysosome (STARD3) (Supplementary Fig. 6e).

We next studied possible STIM1 oligomerization which can occur after ER Ca$^{2+}$ emptying and lead to SOCE activation through Ca$^{2+}$ release-activated channel (CRAC)[28]. Indeed, STIM1 puncta were clearly observed after rotenone or O/A treatment (Supplementary Fig. 7a), supporting ER Ca$^{2+}$ emptying by mitochondrial stressors. Furthermore, STIM1 puncta colocalized with Orai1, suggesting Ca$^{2+}$ influx through SOCE[29] during mitophagy (Supplementary Fig. 7a). To confirm the role of extracellular Ca$^{2+}$ influx through Orai1 in mitophagy, we employed CRAC inhibitors. When we treated cells with BTP2 inhibiting Orai1[30,31], mitophagy by rotenone or O/A was significantly reduced (Supplementary Fig. 7b). When we depleted extracellular Ca$^{2+}$ content using 2 mM EGTA reducing [Ca$^{2+}$]$_i$ approximately to 161 nM (https://somapp.ucdmc.ucdavis.edu/pharmacology/bers/maxchelator/CaEGTA-TS.htm)[32], mitophagy by rotenone or O/A was again significantly reduced (Supplementary Fig. 7b), supporting the importance of extracellular Ca$^{2+}$ influx in mitophagy. [Ca$^{2+}$]$_{Lys}$ recovery after mitochondrial stressor removal was also reduced by incubation in a Ca$^{2+}$-free KRB buffer or 2 mM EGTA (Supplementary Fig. 6a), suggesting the role of extracellular Ca$^{2+}$ influx in ER → lysosome Ca$^{2+}$ refilling.

**Role of *Tfeb* and mitophagy in β-cell function.** We next studied in vivo role of TFEB-mediated mitophagy in pancreatic β-cell function. To this end, we generated *Tfeb*$^{Δβ\text{-cell}}$ mice by crossing *Tfeb*-floxed (*Tfeb*$^{F/F}$) mice to RIP-*Cre* mice, and imposed metabolic stress by feeding HFD. In this model, we wanted to confirm that metabolic stress can cause mitochondrial stress leading to

mitophagy. We thus treated INS-1 cells with palmitic acid (PA), the most abundant saturated fatty acid in vivo and an effector of insulin resistance or β-cell failure[33,34]. PA treatment of INS-1 cells induced mitochondrial stress, as shown by mitochondrial ROS accumulation (Fig. 4a). PA treatment of INS-1 cells also induced a decrease of [Ca$^{2+}$]$_{Lys}$ and an increase of [Ca$^{2+}$]$_i$ (Fig. 4b, c), suggesting that PA-induced mitochondrial ROS influences lysosomal Ca$^{2+}$ channel. Lysosomal pH was not changed by PA treatment of INS-1 cells (Supplementary Fig. 2b), suggesting that PA-induced [Ca$^{2+}$]$_{Lys}$ decrease was not due to lysosomal pH change. Furthermore, TFEB nuclear translocation and mitophagy detected as acidic Mito-Keima puncta were observed after PA treatment (Fig. 4d, e), probably due to TFEB activation by increased [Ca$^{2+}$]$_i$. PA-induced TFEB nuclear translocation was significantly attenuated by NAC or Mito-TEMPO (Fig. 4f), suggesting the role of mitochondrial ROS in PA-induced TFEB activation. In contrast, autophagic activity represented as red puncta after *mRFP-GFP-LC3* transfection was not increased by PA (Supplementary Fig. 2c), which is consistent with previous results[35,36] and suggests that changes of intracellular milieu and associated cellular responses after mitochondrial stressor treatment might affect flux of autophagy or selective autophagy differently. ER-phagic flux represented as red puncta after *mCherry-GFP-RAMP4* transfection[37] was also not increased but decreased by PA (Supplementary Fig. 2d), consistent with a previous paper[38].

While these results suggested the role of PA-induced mitochondrial ROS in mitophagy, PA can affect other signaling pathway such as fatty acid receptors belonging to G protein-coupled receptors (GPCRs) family[39] or can be converted to other lipid intermediates through palmitoyl-CoA or sphingosine/ceramide pathway[40]. We thus examined the effect of inhibitors of such pathways. Pretreatment with DC260126 (a GPCR antagonist)[39], Triacsin C (an acyl-CoA synthetase inhibitor), Fumonisin B1 (a sphingosine *N*-acyltransferase inhibitor) or Myriocin (an inhibitor of serine palmitoyltransferase, thus ceramide synthesis)[40] did not reduce PA-induced mitochondrial ROS accumulation or TFEB nuclear translocation (Supplementary Fig. 8a, b), suggesting that mitochondrial ROS accumulation by PA itself is the most likely event leading to mitophagy.

Since effector molecules of metabolic stress such as PA can induce mitophagy through TFEB activation in vitro, we investigated whether mitophagy occurs in vivo after HFD feeding in pancreatic islet cells expressing *Parkin* (Supplementary Fig. 1c). Indeed, colocalization of LC3, an autophagic marker, with

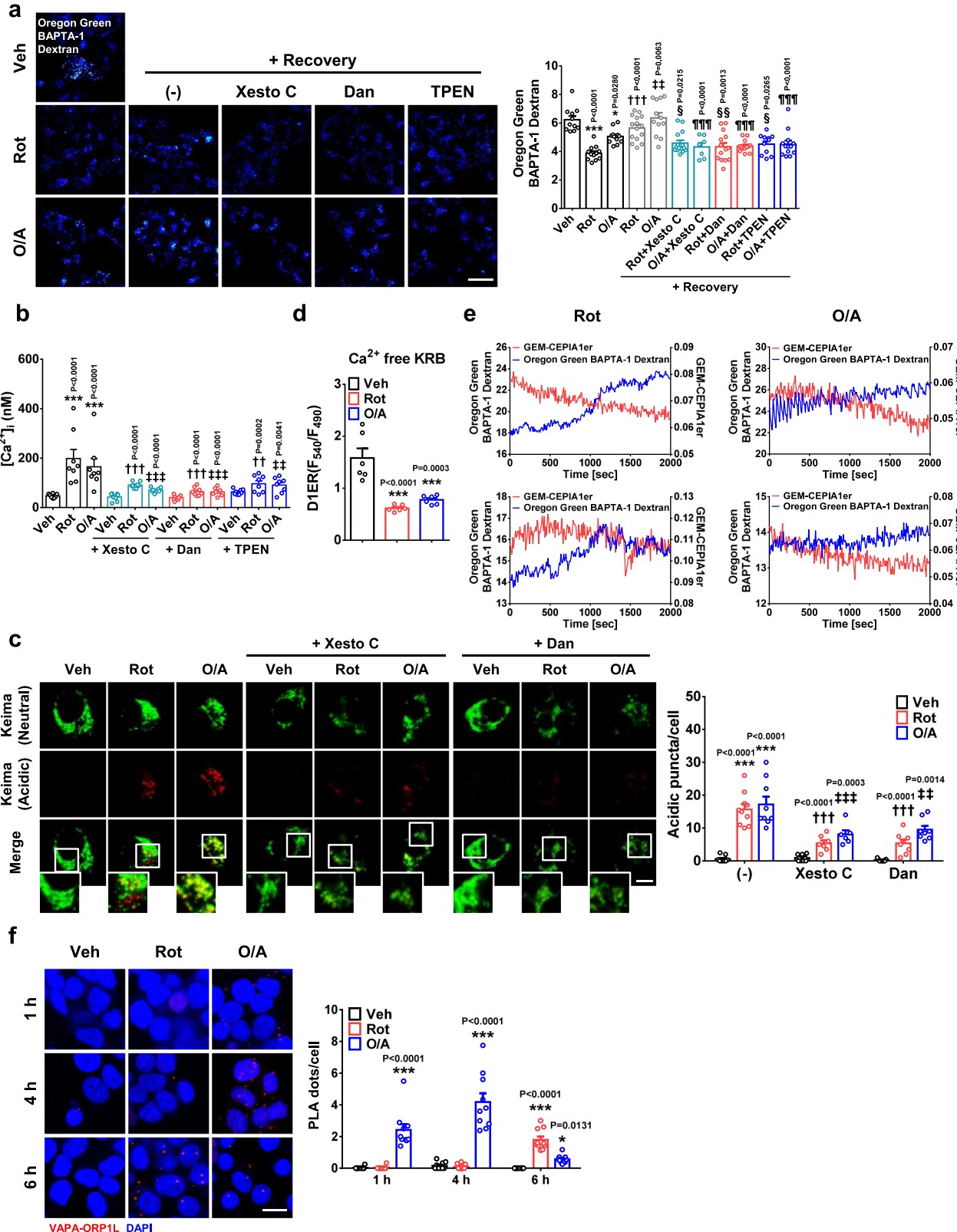

TOM20 in pancreatic islets of $Tfeb^{F/F}$ mice assessed by Pearson's correlation analysis was markedly increased by HFD (Fig. 5a), indicating mitophagy. Intriguingly, increased mitophagy in islets of HFD-fed mice was significantly reduced by β-cell $Tfeb$ KO (Fig. 5a), suggesting TFEB-dependent mitophagy by metabolic stress. As the number of LC3 autophagosomes might not reflect autophagic or mitophagic activity, we studied TOM20

colocalization with a lysosomal marker, as an indicator of mitophagic activity[41]. Mitophagic activity estimated by colocalization of TOM20 with LAMP2 was markedly increased in islets of HFD-fed $Tfeb^{F/F}$ mice. Furthermore, the increased mitophagic activity in islets of HFD-fed mice was significantly reduced by β-cell $Tfeb$ KO (Fig. 5b), suggesting TFEB-dependent increase in β-cell mitophagic activity by HFD. Electron microscopy (EM)

**Fig. 3 ER → lysosome Ca$^{2+}$ refilling during mitophagy. a** Oregon Green 488 BAPTA-1 Dextran-labeled INS-1 cells were treated with mitochondrial stressors for 1 h. After removal of stressors for 1 h, recovery of [Ca$^{2+}$]$_{Lys}$ was determined with and without Xesto C, Dan or TPEN pretreatment (right). Representative fluorescence images are presented (left). (Xesto C, Xesto-spongin C; Dan, Dantrolene) (scale bar, 20 μm) ($n = 7$ for O/A + Recovery + Xesto C; $n = 11$ for Veh or Rot + Recovery + TPEN; $n = 12$ for Rot, O/A or O/A + Recovery; $n = 13$ for O/A + Recovery + Dan or O/A + Recovery + TPEN; $n = 14$ for Rot + Recovery + Dan; $n = 15$ for Rot + Recovery or Rot + Recovery + Xesto C) **b** [Ca$^{2+}$]$_i$ was determined in cells treated with mitochondrial stressors for 1 h with or without Xesto C, Dan or TPEN pretreatment using Fura-2. ($n = 8$) **c** *pMito-Keima*-transfected cells were treated with rotenone or O/A for 18 h with and without Xesto C or Dan pretreatment, and the number of red acidic puncta in cells was determined (right). Representative fluorescence images are presented (left). (scale bar, 5 μm) ($n = 6$ for Xesto C + O/A; $n = 7$ for Xesto C + Rot; $n = 8$ for Dan + Rot or Dan + O/A; $n = 9$ for Veh, O/A, Xesto C or Dan; $n = 10$ for Rot) **d** [Ca$^{2+}$]$_{ER}$ was determined after treating *D1ER*-transfected cells with mitochondrial stressors for 1 h in the absence of extracellular Ca$^{2+}$. ($n = 6$) **e** Cells transfected with *GEM-CEPIA1er* and labeled with Oregon Green 488 BAPTA-1 Dextran were treated with mitochondrial stressors for 1 h. After removal of stressors, the recovery of [Ca$^{2+}$]$_{Lys}$ and changes in [Ca$^{2+}$]$_{ER}$ were determined simultaneously in the absence of extracellular Ca$^{2+}$. **f** Cells were treated with rotenone or O/A, and contact between ER and lysosome was studied by PLA using Abs to VAPA on ER and to ORP1L on lysosome (right). Representative fluorescence images are presented (left). Values on the y-axis represent the numbers of PLA dots in the left panel after treatment with Rot (red bar) or O/A (blue bar). (scale bar, 10 μm) ($n = 10$) Cells in the rectangles were magnified. All data in this figure are the means ± SEM from more than 3 independent experiments. *P* values were determined using one-way ANOVA with Tukey's test. *, compared to Veh-treated cells; †, compared to cells treated with Rot alone; ‡, compared to cells treated with O/A alone; §, compared to Rot-treated cells after recovery; ¶, compared to O/A-treated cells after recovery.

---

also demonstrated decreased aspect ratio of mitochondria[42] in β-cells of HFD-fed *Tfeb*$^{F/F}$ mice, suggestive of mitophagy[43], which was reversed by β-cell *Tfeb* KO (Supplementary Fig. 9a). The number of mitochondria surrounded by isolation membrane or autophagosome in β-cells of HFD-fed *Tfeb*$^{F/F}$ mice was also significantly increased compared to that of normal chow diet (NCD)-fed *Tfeb*$^{F/F}$ or HFD-fed *Tfeb*$^{Δβ-cell}$ mice (Supplementary Fig. 9b). In contrast, autophagic activity determined by colocalization between LAMP2 and LC3[44] was not significantly changed by HFD feeding or *Tfeb* KO (Supplementary Fig. 9c), which is in agreement with no increase of autophagic activity by PA in vitro and increased mitophagic activity without increased autophagic activity in cardiac tissue by HFD for 2 months[45]. The number of autophagosome counted by EM was also not changed by HFD or *Tfeb* KO (Supplementary Fig. 9d).

Probably due to decreased mitophagy, ROS accumulation detected by 4-hydroxynonenal (HNE) staining that was increased in islets of *Tfeb*$^{F/F}$ mice on HFD was further increased by β-cell *Tfeb* KO (Fig. 5c). Regarding mitochondrial activity, cytochrome *c* oxidase (COX) activity in pancreatic islets was increased by HFD probably as an adaptation to metabolic stress, which was significantly reduced by β-cell *Tfeb* KO (Fig. 5d), suggesting TFEB-dependent functional mitochondrial adaptation to metabolic stress. Seahorse respirometry also showed significant increases of basal, glucose-stimulated, ATP-coupled and maximal O$_2$ consumption in pancreatic islets of HFD-fed mice again probably as an adaptation to metabolic stress. These increases of O$_2$ consumption were abolished by *Tfeb* KO (Fig. 5e), supporting TFEB-dependent adaptive response to metabolic stress.

Metabolically, NCD-fed *Tfeb*$^{Δβ-cell}$ mice showed no abnormality in glucose profile up to 20 weeks of age (Supplementary Fig. 10a). However, HFD-fed *Tfeb*$^{Δβ-cell}$ mice showed aggravated glucose intolerance compared to HFD-fed *Tfeb*$^{F/F}$ mice, judged by increased AUC of glucose tolerance test (GTT) (Fig. 6a), while non-fasting blood glucose during or after HFD feeding for 20 weeks did not significantly differ between them (Supplementary Fig. 10a). Aggravated glucose intolerance of HFD-fed *Tfeb*$^{Δβ-cell}$ mice was accompanied by reduced β-cell function compared to HFD-fed *Tfeb*$^{F/F}$ mice as shown by decreased insulinogenic index (Fig. 6b) without significant changes of β-cell mass (Supplementary Fig. 10b), suggesting that *Tfeb*-KO β-cells could not functionally adapt to increased demand for insulin due to insufficient mitophagy and mitochondrial activity, leading to deteriorated glucose intolerance when metabolic stress was imposed. In contrast, AUC of insulin tolerance test (ITT) and body weight did not differ between

*Tfeb*$^{Δβ-cell}$ and *Tfeb*$^{F/F}$ mice on either NCD or HFD (Supplementary Fig. 10c, d).

While these results demonstrated the role of TFEB in β-cell function and metabolic profile, TFE3 that is activated by mitochondrial stress might also play a non-redundant role in β-cell response to metabolic stress. We thus crossed *Tfeb*$^{Δβ-cell}$ mice to *Tfe3*-KO mice. While *Tfeb*$^{F/F}$/*Tfe3*-KO and *Tfeb*$^{Δβ-cell}$/*Tfe3*-KO mice had slightly impaired glucose tolerance compared to *Tfeb*$^{F/F}$/*Tfe3*-WT mice and *Tfeb*$^{Δβ-cell}$/*Tfe3*-WT, respectively, these differences were statistically insignificant in both NCD- and HFD-fed conditions (Supplementary Fig. 10e), suggesting no further independent role of endogenous TFE3 in metabolic adaptation of β-cells.

Using *Tfeb*$^{Δβ-cell}$ mice, we studied which genes are differentially regulated by *Tfeb*. Expression of *Tfeb* downstream genes related to autophagy such as *LC3* (*Map1lc3b*), *Clcn7* and *Ctsa* was upregulated in pancreatic islets of HFD-fed mice together with increased expression of *Tfeb* and *Tfe3* themselves (Fig. 6c). Increased expression of these genes was significantly reduced by β-cell *Tfeb* KO (Fig. 6c), suggesting TFEB-dependent downstream gene induction in pancreatic islets by HFD. Intriguingly, expression of *Ndp52* and *Optn*, crucial autophagy receptors related to mitophagy[46], was significantly increased in pancreatic islets of HFD-fed *Tfeb*$^{F/F}$ mice, and these increases were markedly reduced by β-cell *Tfeb* KO (Fig. 6c), which could be related to TFEB-dependent increase of mitophagy by HFD. Expression of other potential mitophagy receptors, such as *Nbr1*, *Tbk1* or *Tax1bp1*, in pancreatic islets was also increased by HFD, but these increases were statistically insignificant (Fig. 6c). Intriguingly, the expression of mitochondrial biogenesis genes such as *Tfam*, *ESRRα* or *PGC-1α* was increased in pancreatic islets of *Tfeb*$^{F/F}$ mice by HFD. Such induction of mitochondrial biogenesis genes by HFD was abrogated by β-cell *Tfeb* KO (Fig. 6c), which is similar to previous results showing *Tfeb*-dependent induction of mitochondrial biogenesis genes in other tissues[47]. Probably due to increased mitochondrial biogenesis in pancreatic islet cells by HFD, normalized density of TOM20 band in islets of *Tfeb*$^{F/F}$ mice on HFD was not different from that of the same mice on NCD despite significantly increased mitophagy which can reduce mitochondrial proteins including TOM20[4] (Supplementary Fig. 10f).

We next imposed mitochondrial stress on pancreatic islets ex vivo. In rotenone- or O/A-treated islets, mRNA expression of *Ndp52*, *Optn* and other autophagy or lysosomal genes such as *Ctsa*, *Atp6v1h*, *Tfeb* or *Tfe3* was significantly increased (Fig. 6d). Furthermore, increases in *Ndp52*, *Optn*, *Tfeb* and *Atp6v1h*

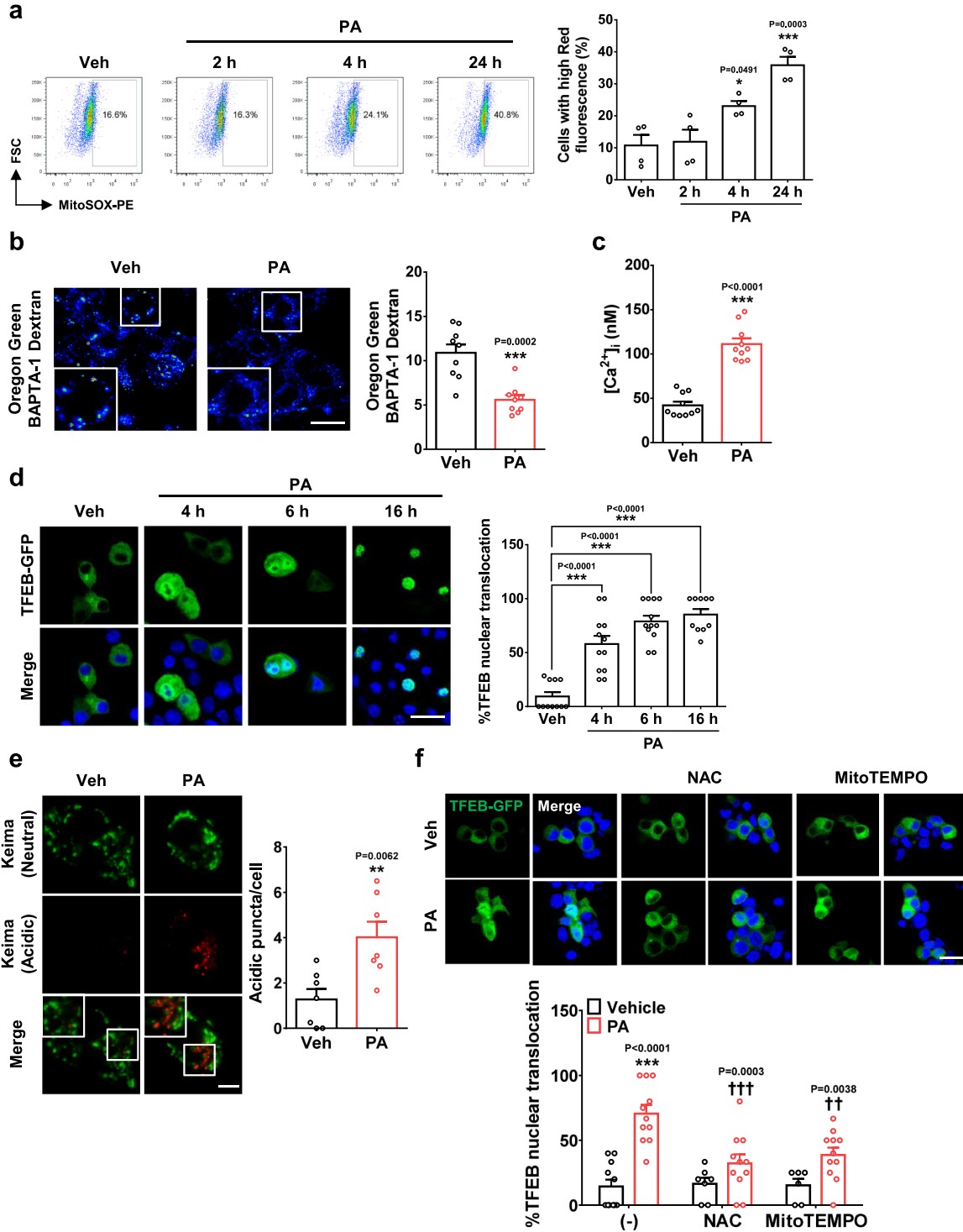

**Fig. 4 Effect of PA on mitochondria and mitophagy. a** After treating INS-1 cells with PA for 2-24 h, mitochondrial ROS was determined using MitoSOX (right). Representative flow cytometric scattergrams are presented (left). ($n = 4$) **b** After treating cells with PA for 1 h, $[Ca^{2+}]_{Lys}$ was determined by Oregon Green 488 BAPTA-1 Dextran loading (right). Representative fluorescence images are presented (left). (scale bar, 20 μm) ($n = 9$) **c** After treating cells with PA for 1 h, $[Ca^{2+}]_i$ was determined using Fura-2 ($n = 10$) **d** After treating *Tfeb-GFP*-transfected cells with PA for 4-16 h, nuclear TFEB was examined by confocal microscopy (right). Representative fluorescence images are presented (left). (scale bar, 20 μm) ($n = 10$ for 16 h; $n = 11$ for Veh; $n = 12$ for 4 h or 6 h) **e** After treating *pMito-Keima*-transfected cells with PA for 18 h, the number of red acidic puncta in transfected cells was determined (right). Representative fluorescence images are presented (left). (scale bar, 5 μm) ($n = 7$) **f** After PA treatment for 4 h with or without ROS quencher pretreatment for 1 h, nuclear TFEB was examined as in (**d**) (lower). Representative fluorescence images are presented (upper). (scale bar, 20 μm) ($n = 6$ for MitoTEMPO; $n = 7$ for NAC; $n = 11$ for Veh, PA, NAC + PA or MitoTEMPO+PA) Cells in the rectangles were magnified. All data in this figure are the means ± SEM from more than 3 independent experiments. *P* values were determined using one-way ANOVA with Tukey's test (**a**, **d**, **f**) or two-tailed *t* test (**b**, **c**, **e**). *, compared to Veh-treated cells; †, compared to cells treated with PA alone.

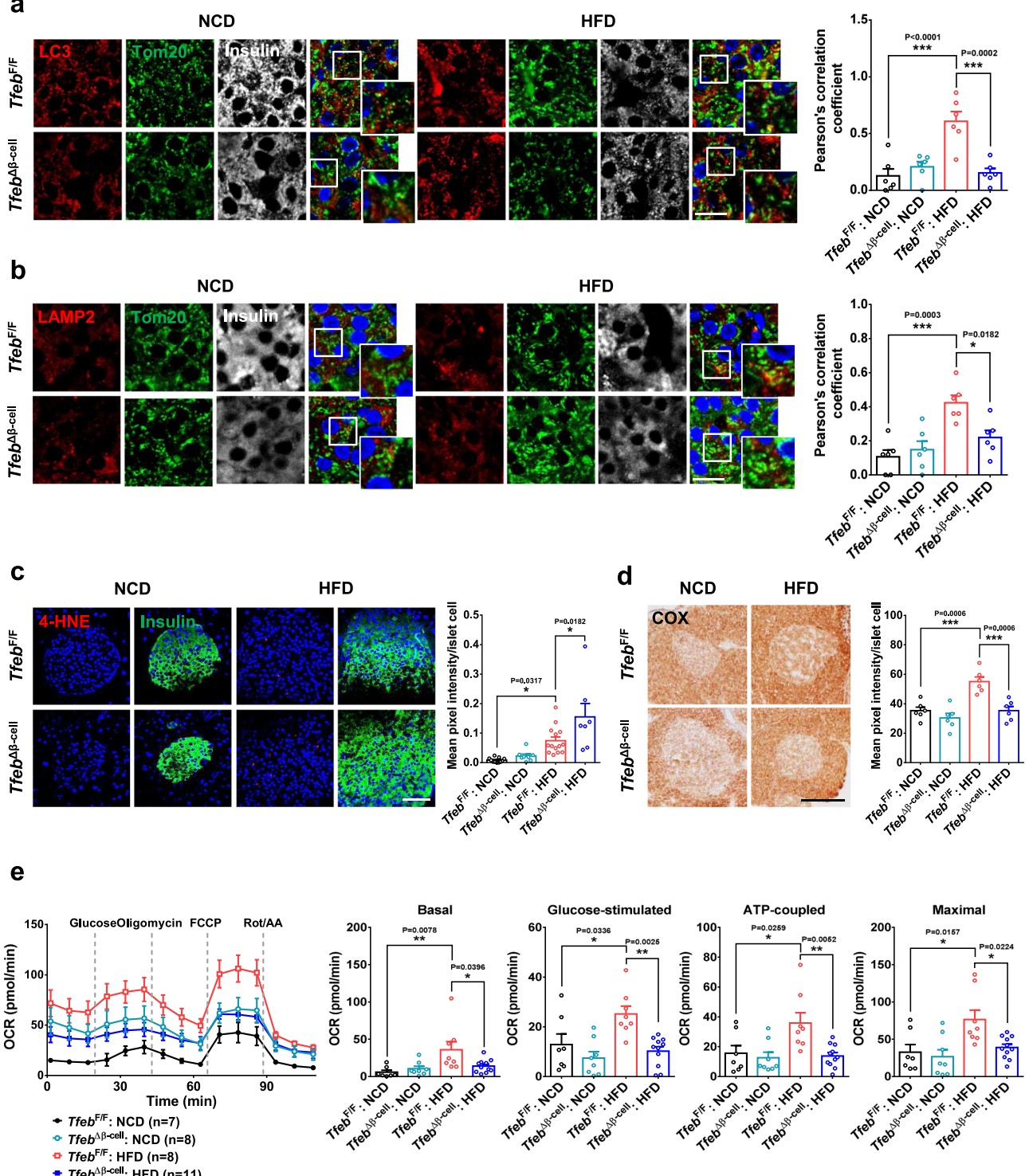

expression by ex vivo rotenone or O/A treatment were significantly reduced by β-cell *Tfeb* KO (Fig. 6d), supporting that TFEB-dependent induction of crucial mitophagy receptors and lysosomal genes by mitochondrial or metabolic stress facilitates mitophagy.

Since these results suggest that *Ndp52* and *Optn* are potential TFEB targets, we conducted chromatin immunoprecipitation (ChIP) assay. To this end, we studied promoters of human *NDP52* and *OPTN* that are well characterized[48]. When 1.1B4 human islet cells generated by electrofusion of human primary pancreatic islet cells to pancreatic adenocarcinoma cells[49] were

treated with rotenone or O/A, N*DP52* or *OPTN* expression was significantly increased, which was downregulated by si*TFEB* (Fig. 6e, f). Contrarily, *TFEB-GFP* transfection significantly increased *NDP52* or *OPTN* expression in 1.1B4 cells (Fig. 6g), supporting the validity of 1.1B4 cells to this end. In rotenone- or O/A-treated 1.1B4 cells, TFEB binding to putative Coordinated Lysosomal Expression and Regulation (CLEAR) sites[50] of *NDP52* or *OPTN* promoter was significantly increased (Fig. 6h), supporting that *NDP52* and *OPTN* are TFEB targets. Since *Nfe2l2* encoding nuclear factor erythroid 2–related factor 2 (Nrf2) can be activated by TFEB or ROS, and *Ndp52* and *Optn* could be

**Fig. 5 Effect of in vivo *Tfeb* KO on mitophagy in β-cells. a** Colocalization of LC3 puncta with TOM20, a mitochondrial marker, in pancreatic islets of *Tfeb*$^{\Delta\beta\text{-cell}}$ and *Tfeb*$^{F/F}$ mice fed NCD or HFD for 12 weeks estimated by Pearson's correlation analysis (right). Representative fluorescence images are presented (left). (scale bar, 10 μm) ($n = 6$) **b** Colocalization of LAMP2 spot, a lysosomal marker, with TOM20 in pancreatic islets of *Tfeb*$^{\Delta\beta\text{-cell}}$ and *Tfeb*$^{F/F}$ mice fed NCD or HFD for 12 weeks estimated by Pearson's correlation analysis (right). Representative fluorescence images are presented (left). (scale bar, 10 μm) ($n = 6$) **c** ROS accumulation determined by HNE staining of pancreatic sections from *Tfeb*$^{\Delta\beta\text{-cell}}$ and *Tfeb*$^{F/F}$ mice fed NCF or HFD for 12 weeks (right). Representative fluorescence images are presented (left). (scale bar, 50 μm) ($n = 7$ for *Tfeb*$^{\Delta\beta\text{-cell}}$:HFD; $n = 8$ for *Tfeb*$^{\Delta\beta\text{-cell}}$:NCD; $n = 11$ for *Tfeb*$^{F/F}$:NCD; $n = 14$ for *Tfeb*$^{F/F}$:HFD) **d** Mitochondrial COX activity in pancreatic islets of *Tfeb*$^{\Delta\beta\text{-cell}}$ and *Tfeb*$^{F/F}$ mice fed NCD or HFD for 12 weeks (right). Representative DAB images are presented (left). (scale bar, 100 μm) ($n = 6$) **e** Seahorse XF respirometry using primary pancreatic islets from *Tfeb*$^{\Delta\beta\text{-cell}}$ and *Tfeb*$^{F/F}$ mice on NCD or HFD for 12 weeks (left). Basal, glucose-stimulated, ATP-coupled and maximal O$_2$ consumptions were calculated from the curves (right). (OCR, O$_2$ consumption rate; FCCP, carbonyl cyanide-4-[trifluoromethoxy]phenylhydrazone; Rot, rotenone; AA, antimycin A) Cells in the rectangles were magnified. All data in this figure are the means ± SEM from more than 3 independent experiments. *P* values were determined using one-way ANOVA with Tukey's test; ns, not significant.

*Nfe2l2* target genes[48,51], we studied the role of *Nfe2l2* in *Ndp52* or *Optn* expression employing pancreatic islets from *Nfe2l2*-KO mice. Induction of *Ndp52* or *Optn* in primary islet cells by rotenone or O/A was not significantly reduced by *Nfe2l2* KO (Fig. 6i), suggesting that mitochondrial stressor-induced *Ndp52* and *Optn* in pancreatic islet cells is largely due to TFEB-mediated transactivation. These results indicate that TFEB-dependent mitophagy, contributed by transactivation of mitophagy receptor genes such as *Ndp52* or *Optn*, plays an important role in β-cell adaptation to metabolic stress through improved mitochondrial function.

Since the induction of *Ndp52* and *Optn* by metabolic or mitochondrial stressors could explain selective mitophagy observed in pancreatic islets of HFD-fed mice, we conducted KD of *Ndp52* and *Optn* in INS-1 cells that show increased expression of *Ndp52* or *Optn* after treatment with PA (Supplementary Fig. 11a). Indeed, KD of *Ndp52* and *Optn* that successfully reduced expression of *Ndp52* and *Optn*, respectively (Supplementary Fig. 11b), significantly reduced mitophagy after PA treatment (Supplementary Fig. 11c), suggesting that induction of *Ndp52* or *Optn* in β-cells contributes to the selective mitophagy after HFD feeding. The role of *Ndp52* and *Optn* in mitophagy after rotenone or O/A treatment was also studied using INS-1 cells that show elevated expression of *Ndp52* and *Optn* after rotenone or O/A treatment (Supplementary Fig. 11d). Mitophagy after rotenone or O/A treatment was also significantly reduced by KD of *Ndp52* or *Optn*. Combined KD of both *Ndp52* and *Optn* further reduced mitophagy after rotenone or O/A treatment (Supplementary Fig. 11e), suggesting that induction of *Ndp52* or *Optn* contributes to mitophagy after treatment with mitochondrial stressors.

## Discussion

Previous studies have shown the role of *Tfeb* family in systemic metabolism[47,52]. However, the role of β-cell *Tfeb* in β-cell function and systemic metabolism has not been studied. Furthermore, while the role of autophagy in β-cell function and metabolism has been studied extensively, the role of mitophagy in β-cell function has been hardly investigated.

Mitophagy of pancreatic β-cells appears to be important in mitochondrial integrity and functional adaptation during metabolic or mitochondrial stress because HFD feeding increased insulinogenic index in wild-type mice but not in *Tfeb*$^{\Delta\beta\text{-cell}}$ mice that showed defective mitophagic response. Although the role of other types of selective autophagy or bulk autophagy modulated by TFEB cannot be disregarded in TFEB-mediated protection of β-cell function, flux of autophagy or ER-phagy was not increased by PA or HFD feeding. These results are consistent with previous papers showing no increase in fluxes of autophagy or ER-phagy in vitro and in vivo by PA or HFD feeding[36,38,45], which could be due to different intracellular environment

and associated cellular responses imposed by diverse organelle stressors such as induction of mitophagy receptors, modifying TFEB-dependent or -independent autophagic activity.

TFEB-mediated mitophagy appears to be important in adaptive mitochondrial respiration because HFD-induced upregulation of mitophagy, COX activity, and mitochondrial O$_2$ consumption in β-cells was abolished by *Tfeb* KO. Increases in mitophagy, COX activity, and mitochondrial O$_2$ consumption in pancreatic islets of HFD-fed mice are in line with previous results reporting increased mitochondrial respiration, complex activities, or content and mitophagy in the heart or islet tissues after HFD feeding[45,53–55], which could be due to fatty acid-induced uncoupling, O$_2$ shunting to ROS or adaptive changes to metabolic stress[54]. Probably due to impaired mitophagy and mitochondrial adaptation to metabolic stress, HFD-induced ROS accumulation was augmented by β-cell *Tfeb* KO, leading to significantly suppressed β-cell function and aggravated glucose intolerance. Consistent with HFD-induced increases of mitophagy and COX activity, induction of TFEB downstream genes or *Tfeb* and *Tfe3* themselves was observed in pancreatic islets of HFD-fed mice, again supporting the role of TFEB activation in adaptation to HFD. However, we observed no further metabolic deterioration by additional *Tfe3* KO despite comparable TFE3 activation by mitochondrial stressors, suggesting no non-redundant role of *Tfe3* in β-cell adaptation to metabolic stress. These results are in contrast to a previous paper showing an independent role of *Tfe3* in metabolic adaptation[56]. Such discrepancies could be due to different types of tissues analyzed and non-identical experimental methods or mouse background.

Although these results suggest important roles of mitophagy in pancreatic β-cell function, previous papers have reported no significant decrease of glucose-stimulated insulin release from β-cells with targeted disruption of *Parkin*[57] or *PINK1*, an upstream kinase recruiting Parkin onto dysfunctional or damaged mitochondria[46,58]. This discrepancy could be attributable to compensatory changes in *Parkin*- or *PINK1*-KO β-cells which might include Parkin-independent mitophagy such as piecemeal-type mitophagy or receptor-mediated mitophagy that can occur in *Parkin*-KO or *PINK1*-KO β-cells[59,60]. In contrast, *Parkin*-KD INS-1 cells showed diminished mitochondrial function and glucose-stimulated insulin release[61], which might be due to the absence of compensatory changes associated with *Parkin* KO. The role of β-cell Parkin in the maintenance of mitochondrial function and glucose-stimulated insulin release after treatment with streptozotocin, a classical β-cell toxin, has also been reported[62], suggesting the role of Parkin-mediated mitophagy of β-cells in stressed condition.

We observed *Ndp52* and *Optn* induction in pancreatic islets by metabolic or mitochondrial stress, which was attenuated by *Tfeb*

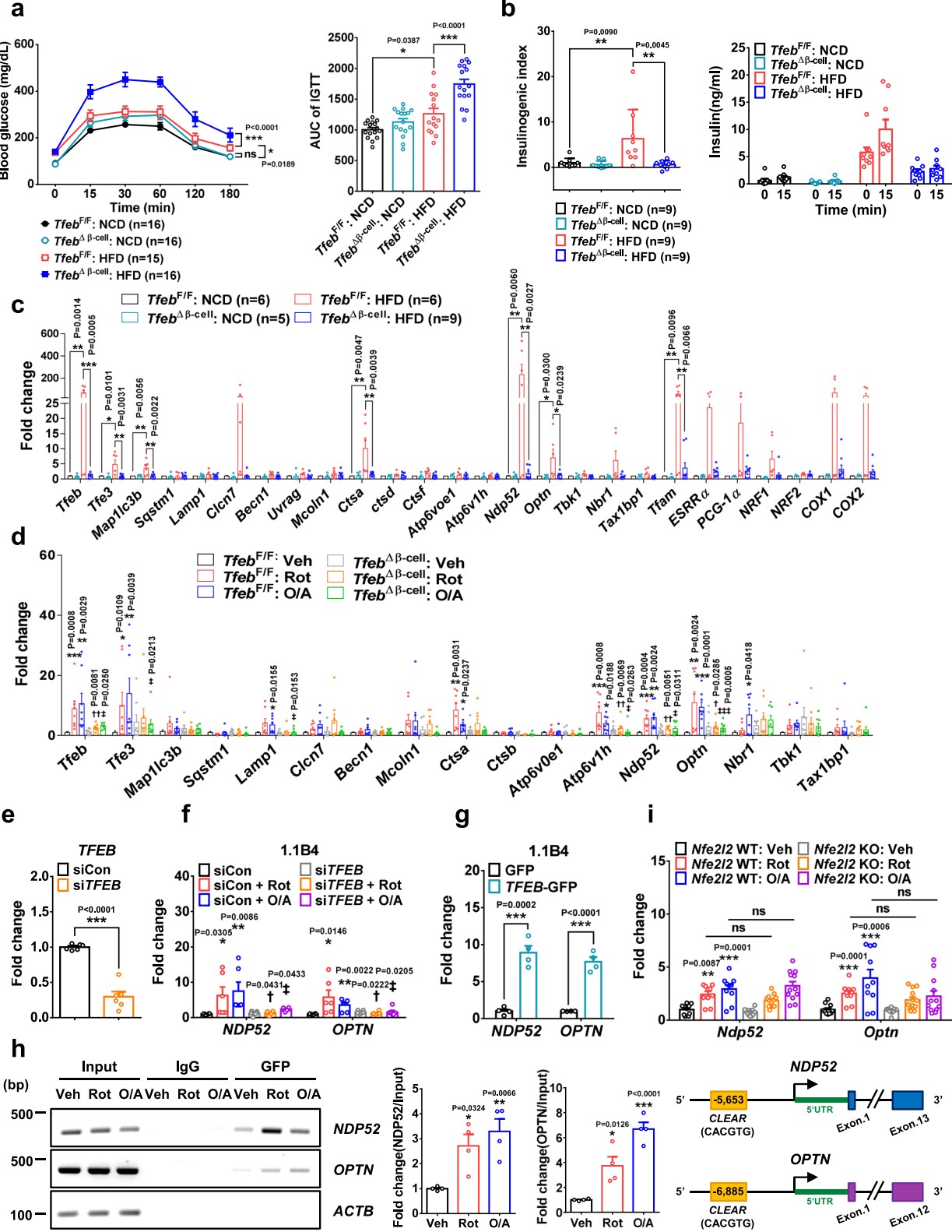

KO. Since NDP52 and OPTN are primary receptors for mitophagy[46], TFEB-dependent induction of *Ndp52* and *Optn* in pancreatic islet cells is likely to contribute to the increased mitophagy by metabolic stress or mitochondrial stress. These results suggest a role of transactivation of *Ndp52* and *Optn* in mitochondrial stress-induced mitophagy besides mitophagy receptor phosphorylation[63].

When we studied the mechanism of mitophagy in β-cells undergoing mitochondrial stress, we observed significantly increased $[Ca^{2+}]_i$ by rotenone or O/A, similar to results using other cell types[64,65]. As antimycin A had a relatively weak effect on mitochondrial potential, it has been used in combination with oligomycin inhibiting reverse activity of $F_1F_0$-ATPase[66]. $Ca^{2+}$ channel responsible for lysosomal $Ca^{2+}$ release by mitochondrial

**Fig. 6 Effect of β-cell-specific *Tfeb* KO on metabolic profile and target gene expression. a** GTT in *Tfeb*^Δβ-cell^ and *Tfeb*^F/F^ mice on NCD or HFD for 12 weeks (left). AUC (right). **b** Insulinogenic index in *Tfeb*^Δβ-cell^ and *Tfeb*^F/F^ mice on NCD or HFD for 12 weeks (left). Serum insulin levels before and 15 min after glucose injection (right). **c** mRNA from pancreatic islets of *Tfeb*^Δβ-cell^ and *Tfeb*^F/F^ mice fed NCD or HFD for 12 weeks was subjected to real-time RT-PCR. **d** Pancreatic islets from NCD-fed *Tfeb*^Δβ-cell^ and *Tfeb*^F/F^ mice were treated with mitochondrial stressors for 6 h. mRNA from the treated islets was subjected to real-time RT-PCR. (*n* = 8) **e** Expression of *TFEB* mRNA in 1.1B4 cells transfected with *TFEB* siRNA examined by real-time RT-PCR. (*n* = 6) **f** After treating *TFEB* siRNA-transfected 1.1B4 cells with mitochondrial stressors for 6 h, expression of the indicated genes was examined by real-time RT-PCR. (*n* = 6) (SiCon, control siRNA) **g** Forty-eight h after transfection of 1.1B4 cells with control *GFP* or *TFEB-GFP*, expression of the indicated genes was examined by real-time RT-PCR. (*n* = 4) **h** After treatment of 1.1B4 cells with mitochondrial stressors, ChIP assay was conducted. Fold changes (middle). Representative immunoblots (left). (*n* = 4) Positions of putative *TFEB*-binding CLEAR sequence in the promoter regions of *NDP52* and *OPTN* (right). **i** Pancreatic islets from *Nfe2l2*-KO and control mice were treated with mitochondrial stressors for 6 h. Expression of *Ndp52* and *Optn* evaluated by real-time RT-PCR mRNA. (*n* = 10 for *Nfe2l2* WT; *n* = 12 for *Nfe2l2* KO) All data in this figure are the means ± SEM from more than 3 independent experiments. *P* values were determined using o one-way ANOVA with Tukey's test (**a–d**, **f**, **h**, **i**), two-way ANOVA with Bonferroni's test (**a**) or two-tailed *t* test (**e**, **g**). ns, not significant; *, compared to (wild-type) cells or control transfectants treated with Veh; †, compared to wild-type cells or control transfectants treated with Rot alone; ‡, compared to wild-type cells or control transfectants treated with O/A alone.

ROS appears to be TRPML1. Besides PI(3,5)P$_2$, ROS can also activate TRPML1 channel but not TRPML2 or TRPML3 channel[5]. Since ML-SI3 is not specific for TRPML1, ML-SI3-mediated abrogation of mitochondrial stressor-induced $[Ca^{2+}]_i$ increase does not prove the role of TRPML1 in TFEB activation and mitophagy by mitochondrial stressors. However, lysosomal $Ca^{2+}$ release by mitochondrial ROS that does not activate TRPML2 or TRPML3 channel and abrogation of mitochondrial stressor-induced TFEB activation by si*Mcoln1* strongly support the role of TRPML1 in mitochondrial stressor-induced lysosomal $Ca^{2+}$ release. Following lysosomal $Ca^{2+}$ release through TRPML1, increased $[Ca^{2+}]_i$ dephosphorylated TFEB and TFE3 via calcineurin activation. We also observed notably decreased total TFEB and TFE3 following O/A treatment. Although the mechanism of this phenomenon is unknown, it could be due to changes in the phosphorylation of amino acid residues related to the stability of TFEB family members[67].

Although the initial lysosomal $Ca^{2+}$ release occurs in response to mitochondrial ROS, small lysosomal $Ca^{2+}$ reservoir might not be sufficient to accomplish full progression of mitophagy[6]. While $[Ca^{2+}]_{Lys}$ is comparable to $[Ca^{2+}]_{ER}$, lysosome is not a major $Ca^{2+}$ reservoir because its volume is 1/10 of ER volume[6]. Previous papers demonstrated occurrence of ER → lysosome $Ca^{2+}$ refilling for completion of events requiring lysosomal $Ca^{2+}$[20]. ER $Ca^{2+}$ channels such as InsP3R or ryanodine receptor channel could act as ER $Ca^{2+}$ exit channels in ER → lysosome $Ca^{2+}$ refilling. Indeed, genetic studies have shown the participation of InsP3R in mitophagy[68]; however, the role of ER $Ca^{2+}$ or InsP3R in lysosomal $Ca^{2+}$ refilling and the relationship between ER $Ca^{2+}$ and lysosomal $Ca^{2+}$ in mitophagy have not been studied. Our data show that ER → lysosome $Ca^{2+}$ refilling occurs during mitochondrial stress, which is necessary for replenishment of lysosomal $Ca^{2+}$ reservoir and completion of TFEB activation or mitophagy.

ER → lysosome $Ca^{2+}$ refilling could be facilitated by ER-lysosome contact[25]. In addition to well-characterized ER-mitochondrial contact, ER-lysosome contact could have a functional role in cellular responses[69]. When ER $Ca^{2+}$ emptying occurs due to ER → lysosome flux[6], SOCE is likely to occur through CRAC activation, which is characterized by STIM1 oligomerization and translocation to ER-plasma junction where it interacts with Orai channels[28]. Thus, STIM1 oligomerization by mitochondrial stressors strongly supports that ER $Ca^{2+}$ emptying occurs due to ER → lysosome $Ca^{2+}$ flux, which also can explain the suppression of mitophagy by extracellular $Ca^{2+}$ depletion[70].

Altogether, these results suggest that TFEB is activated by lysosomal $Ca^{2+}$ release coupled with ER → lysosome $Ca^{2+}$ refilling during mitochondrial or metabolic stress, and TFEB-dependent induction of mitophagy, facilitated by transactivation of mitophagy receptors such as *Ndp52* or *Optn*, plays an important role in β-cell adaptation to metabolic stress through improved mitochondrial function.

## Methods

**Detection of mitophagy in vitro**. To visualize mitophagy using a mitophagy-specific probe, INS-1 cells were transfected with a *pMito-Keima* (MBL), and then treated with rotenone or O/A for 18 h. Keima fluorescence at neutral pH was observed at an excitation wavelength of 430 ± 20 nm and an emission wavelength of 624 ± 20 nm. Keima fluorescence in acidic conditions was observed at an excitation wavelength of 562 ± 20 nm and emission wavelength of 624 ± 20 nm. To visualize mitophagy in another way, cells transfected with the *mRFP-LC3* construct were treated with mitochondrial stressors and then incubated with anti-TOM20 Ab (Cell Signaling Technology, 1:200). Images were obtained using an LSM780 confocal microscope (Zeiss) from more than five different fields for each experiment. Colocalization of puncta or fluorescence in vitro was evaluated manually in a blinded manner.

**GCaMP3 $Ca^{2+}$ imaging**. INS-1 cells were transfected with a plasmid encoding a perilysosomal GCaMP3-ML1 $Ca^{2+}$ probe[12]. After 48 h, lysosomal $Ca^{2+}$ release was measured in a zero $Ca^{2+}$ solution (145 mM NaCl, 5 mM KCl, 3 mM MgCl$_2$, 10 mM glucose, 1 mM EGTA and 20 mM HEPES, pH 7.4) with or without rotenone or O/A pretreatment for 1 h by monitoring fluorescence intensity at 470 nm with a LSM780 confocal microscope (Zeiss). GPN was added at the indicated time.

**Cytosolic $Ca^{2+}$ measurement**. For ratiometric measurement of $[Ca^{2+}]_i$, cells were loaded with 2 μM of the acetoxymethyl ester form of Fura-2 (Invitrogen) in RPMI-1640 at 37 °C for 30 min and then rinsed with fresh solution. Measurements were performed using MetaFluor (Molecular devices) on an Axio Observer A1 (Zeiss) equipped with a 150 W xenon lamp Polychrome V (Till Photonics), a CoolSNAP-Hq2 digital camera (Photometrics), and a Fura-2 filter set. $[Ca^{2+}]_i$ was determined in a phenol red-free RPMI-1640 using the 340/380 nm fluorescence ratio and the following equation[71].

$$[Ca^{2+}]_i = K_d \times [(R - R_{min})/(R_{max} - R)] \times [F_{min(380)}/F_{max(380)}] \quad (1)$$

where $K_d$ = Fura-2 dissociation constant (224 nM at 37 °C), $F_{min(380)}$ = the 380 nm fluorescence in the absence of $Ca^{2+}$, $F_{max(380)}$ = 380 nm fluorescence with saturating $Ca^{2+}$, $R$ = 340/380 nm fluorescence ratio, $R_{max}$ = 340/380 nm ratio with saturating $Ca^{2+}$, and $R_{min}$ = 340/380 nm ratio in the absence of $Ca^{2+}$.

To monitor $[Ca^{2+}]_i$ after GPN treatment as an estimate of lysosomal $Ca^{2+}$ content, cells were incubated in a zero $Ca^{2+}$ solution containing 145 mM NaCl, 5 mM KCl, 3 mM MgCl$_2$, 10 mM glucose, 1 mM EGTA and 20 mM HEPES (pH 7.4)[20].

**Measurement of lysosomal $Ca^{2+}$ content ($[Ca^{2+}]_{Lys}$)**. To measure $[Ca^{2+}]_{Lys}$, cells were loaded with 100 μg/mL of Oregon Green 488 BAPTA-1 Dextran (Invitrogen) at 37 °C for 12 h. Medium was then changed to one without Oregon Green 488 BAPTA-1 Dextran, and cells were chased for an additional 4 h. After treating cells with rotenone or O/A for the last 1 h of chasing, cells were washed in HEPES-buffered saline (135 mM NaCl, 5.9 mM KCl, 1.2 mM MgCl$_2$, 1.5 mM CaCl$_2$, 11.5 mM glucose, 11.6 mM HEPES, pH 7.3) for confocal microscopy[20].

**Measurement of ER $Ca^{2+}$ content ($[Ca^{2+}]_{ER}$)**. INS-1 cells were transfected with *GEM-CEPIA1er* (Addgene)[23] or a ratiometric FRET-based Cameleon probe *D1ER*[24] using Lipofectamine 2000. After 48 h, cells were treated with rotenone or O/A in a $Ca^{2+}$-free KRB buffer (Sigma) for 1 h to inhibit influx of extracellular $Ca^{2+}$ into ER[34] or in a culture medium for 1 h. GEM-CEPIA1er fluorescence was

measured using an LSM780 confocal microscope (Zeiss) at an excitation wavelength of 405 nm and emission wavelengths of 466 or 520 nm, and F466/F520 was calculated as an index of $[Ca^{2+}]_{ER}$[23]. D1ER fluorescence intensity ratio (F540/F490) was determined using an LSM980 confocal microscope (Zeiss)

**Simultaneous monitoring of $[Ca^{2+}]_{Lys}$ and $[Ca^{2+}]_{ER}$.** Twenty-four h after transfection with GEM-CEPIA1er, INS-1 cells were loaded with Oregon Green 488 BAPTA-1 Dextran for 12 h and chased for 4 h. After treating cells with rotenone or O/A for 1 h, cells were monitored for $[Ca^{2+}]_{Lys}$ and $[Ca^{2+}]_{ER}$ in a $Ca^{2+}$-free KRB buffer in the presence or absence of rotenone or O/A using an LSM780 confocal microscope.

**Lysosomal pH.** After incubating cells with mitochondrial stressors or PA and 1 μM LysoSensor green DND-189 in a pre-warmed HBSS buffer (135 mM NaCl, 5.9 mM KCl, 1.2 mM MgCl₂, 1.5 mM CaCl₂, 11.5 mM glucose, 11.5 mM HEPES, pH 7.3) for 10 min in 37 °C incubator, LysoSensor green intensity was acquired using a confocal microscope and converted to pH using a standard curve obtained after treating cells with 10 μM nigericin for 20 min in 105 mM KCl buffer (105 mM KCl, 25 mM HEPES, 1 mM MgCl₂) adjusted to different pH[72].

**Proximity ligation assay (PLA).** Contact between ER and lysosome was studied using a Duolink InSitu Detection Reagents Red kit (Sigma) according to the manufacturer's protocol. Briefly, cells were incubated with primary Abs against ORP1L (Abcam, 1:200) and VAPA (Santa Cruz, 1:200) at 4 °C overnight. After washing, cells were incubated with PLA plus and minus probes at 37 °C for 1 h. After ligation reaction to close the circle and rolling circle amplification (RCA) of the ligation product, fluorescence-labeled oligonucleotides hybridized to the RCA products were observed by fluorescence microscopy.

**Contact-ID.** Full-length DNA fragments of mouse VAPA and STARD3 were produced from cDNA from J774A.1 (ATCC, TIB-67) cells by PCR. DNA fragments encoding VAPA and STARD3 were inserted into Age1-XhoI site of Flag-BirA(N-G78) and Hind3-BamH1 site of BirA(G79-C)-HA vectors, respectively, using T4 DNA ligase (Takara). Nucleotide sequences of all constructs were confirmed by automatic DNA sequencing. Twenty-four h after transfection of INS-1 cells with Flag-BirA-VAPA and STARD3-BirA-HA using Lipofectamine 2000, 50 μM biotin was added for labeling overnight. After treatment with rotenone or O/A for 1 h, cells were fixed with 4% paraformaldehyde and then permeabilized with 0.5% Triton X-100-PBS. After 10% goat serum blocking, cells were sequentially incubated with anti-FLAG (Sigma, 1:200) and -HA Abs (Cell Signaling Technology, 1:200) diluted in PBS-1% BSA, secondary Abs diluted in PBS-1% BSA for 30 min, and then with FITC-streptavidin diluted in PBS-1% BSA for 30 min at room temperature. (VAPA-Age1 forward: CGACCGGTCATGGCGTCCGCCTCC GGG GCC ATG GC; VAPA-Xho1 reverse: GCCTCGAGCTACAAGATGAATT TCCCTAGAAAGAA; STARD3-Hind3 forward: CCAAGCTTATGAGCAAGCGA CCTGGTGATCTGGCC; STARD3-BamH1 reverse: CGGATCCAGCTCGGGCC CCCAGCTCTCCAACACG).

**CRAC activation.** INS-1 cells transfected with YFP-STIM1 and 3 x FLAG-mCherry Red-Orai1/P3XFLAG7.1 (provided by Yuan J through Cha S-G) were treated with mitochondrial stressors for 30 min and then subjected to confocal microscopy to visualize STIM1 puncta and their colocalization with Orai1. Colocalization between STIM1 and Orai1 was confirmed by calculating Pearson's correlation coefficient.

**Cell culture and drug treatment.** INS-1 cells (kindly provided by Wollheim C) were cultured in RPMI-1640 containing 11.2 mM glucose and 2 mM glutamine supplemented with 10% FBS, 1 mM pyruvate, 10 mM HEPES, 50 μM 2-mercaptoethanol, 100 U/mL penicillin and 100 μg/mL streptomycin. INS-1 cells were seeded onto wells of Poly-L-ornithine-coated plates. 1.1B4 cells obtained from ECACC (Salisbury) through Abdul Majid FA were cultured in RPMI-1640-10% FBS, 100 U/mL penicillin and 100 μg/mL streptomycin. For drug treatment, the following concentrations were used: 100 nM rotenone (Sigma), 200 nM oligomycin (Millipore), 125 nM antimycin A (Sigma), 400 μM PA (Sigma), 50 μM BAPTA-AM (Invitrogen), 20 μM ML-SI3, 200 μM GPN (Sigma), 3 μM Xesto-spongin C (Abcam), 10 μM Dantrolene (Sigma), 10 μM TPEN (Sigma), 10 μM BTP2 (Sigma), 2 mM EGTA (Sigma), 5 mM N-acetylcysteine (Sigma) and 100 μM MitoTEMPO (Sigma), 2 μM Thapsigargin (Sigma), 10 μM DC260126 (Cayman), 10 μM Fumonisin B1 (Sigma), 10 μM Myriocin (Sigma) and 5 μM Triacsin C (Abcam). PA stock solution (50 mM) was prepared by dissolving in 70% ethanol and heating at 55 °C[40]. The working solution was made by diluting PA stock solution in 2% fatty acid-free BSA-RPMI 1640. To impose mitochondrial stress on primary pancreatic islets, higher doses of mitochondrial stressors (1 μM rotenone, 2 μM oligomycin, and 1.25 μM antimycin A) were employed.

**Transfection and plasmids.** Cells were transiently transfected with the following plasmids: MT-Keima, TFEB-GFP, TFE3-GFP, mRFP-LC3, mRFP-GFP-LC3, pcDNA3.1-HA, pcDNA3.1-HA-ΔCnA-H151Q, GCaMP3-ML1, GEM-CEPIA1er, YFP-STIM1, D1ER or mCherry-Orai1 using Lipofectamine 2000 (Invitrogen)

according to the manufacturer's protocol. In the case of TetOn-mCherry-GFP-RAMP4 (Addgene), cells were treated with 4 μg/ml Doxycycline for 48 h after transfection for 24 h before use. When HA-ΔCnA-H151Q was used, the same dose of regulatory calcineurin B subunit was transfected together. To evaluate autophagic and ER-phagic fluxes, the numbers of red puncta of transfected mRFP-GFP-LC3 and TetOn-mCherry-GFP-RAMP4[37] were counted, respectively.

**Gene knockdown (KD).** INS-1 cells were transfected with Mcoln1 siRNA (CCCACAUCCAGGAGUGUAA), Ndp52 siRNA (CACACAGCAGAA AAUCCAA), Optn siRNA (CAGCUUACAGAUCUCAGUU) or control siRNA using Lipofectamine RNAiMAX (Invitrogen), according to the manufacturer's protocol. 1.1B4 cells were transfected with TFEB siRNA (UGUAAUGCAUGA CAGCCUG) or control siRNA using Lipofectamine RNAiMAX. Efficacy of Mcoln1 or TFEB siRNA KD was evaluated by RT-PCR. Primer sequences are listed in the Supplementary Tables 1, 3.

**Abs and immunoblot analyses.** Cells were solubilized in a lysis buffer containing protease inhibitors. Protein concentration was determined using the Bradford method. Samples (10–30 μg) were separated on 4–12% Bis-Tris gel (NUPAGE, Invitrogen) and transferred to nitrocellulose membranes for immunoblot analysis using the ECL method (Dongin LS). For immunoblotting, Abs against the following proteins were used: Parkin (Santa Cruz, 1:1,000), TFEB (Bethyl Laboratories, 1:1,000), TFE3 (Sigma Aldrich, 1:1,000), phospho-S142-TFEB (Millipore, 1:1,000), phospho-(Ser) 14-3-3 binding motif (Cell Signaling Technology, 1:1,000), HSP70 (Santa Cruz, 1:1,000), Tom20 (Cell Signaling Technology, 1:1,000), HSP90 (Abcam, 1:1,000) and ACTB (Santa Cruz, 1:1,000).

**Immunoprecipitation.** After lysis of Tfeb- or Tfe3-GFP-transfected INS-1 cells in an ice-cold lysis buffer (400 mM NaCl, 25 mM Tris-HCl, pH 7.4, 1 mM EDTA, and 1% Triton X-100) containing protease and phosphatase inhibitors, lysates were centrifuged at 12,000 g for 10 min in microfuge tubes, and supernatant was incubated with anti-GFP (AbFrontier, 1:1,000) in binding buffer (200 mM NaCl, 25 mM Tris-HCl, pH 7.4, 1 mM EDTA) with constant rotation at 4 °C for 1 h. After the addition of 50 μL of 50% of Protein-G bead (Roche Applied Science) slurry to lysates and incubation with rotation at 4 °C overnight, resins were washed with binding buffer. After resuspending pellet in a sample buffer (Life Technology) and heating at 100 °C for 3 min, supernatant was collected by centrifugation at 12,000 g for 30 sec for electrophoretic separation in a NuPAGE® gradient gel (Life Technology). Immunoblotting was conducted by sequential incubation with anti-phospho-(Ser) 14-3-3 binding motif Ab (Cell Signaling Technology, 1:1,000) as the primary Ab and horseradish peroxidase-conjugated anti-rabbit IgG. Bands were visualized using an ECL kit for the detection of chemiluminescence.

**Immunofluorescence.** For TFEB and TFE3 immunofluorescence, cells treated with mitochondrial stressors were fixed with 4% paraformaldehyde and permeabilized with 0.5% Triton X-100. Cells were then blocked with 10% bovine serum albumin in PBS. After sequential incubation of cells with anti-TFEB (Bethyl Laboratories, 1:200) or anti-TFE3 Ab (Sigma, 1:200) at 4 °C overnight and then with anti-rabbit secondary Abs conjugated to Alexa Fluor 488 (Invitrogen) for 1 h, fluorescence was visualized with an LSM780 confocal microscope (Zeiss).

**β-cell mass.** Relative β-cell mass was determined by point counting morphometry after insulin immunohistochemistry using anti-insulin Ab (Cell Signaling Technology, 1:200)[9]. The number of test points was selected using a nomogram to set the probable error of volume density to less than 10% of the calculated value.

**Electron Microscopy (EM).** After fixation in 2.5% glutaraldehyde-0.1 M NaH₂PO₄/Na₂HPO₄ phosphate buffer (pH 7.2) and phosphate buffer washing, pancreas tissues were post-fixed with 1% OsO₄ and embedded in Epon812. Ultrathin sections were mounted and stained with uranyl acetate/lead citrate (JEM-1011, Jeol). Aspect ratio was calculated as the ratio between the length of the major axis and that of the minor axis of mitochondria[42]. The number of mitochondria surrounded by isolation membrane or autophagosome was counted in 7~8 fields of magnified images (x 10,000) per mouse for a total of 25 fields as an index of mitophagy. The number of double-membrane structure with a luminal density similar to the cytoplasm but without identifiable organelle structure representing bulk autophagy was also counted in the same fields.

**RNA extraction and real-time RT-PCR.** Total RNA was extracted using TRIzol (Invitrogen), and cDNA was synthesized using M-MLV Reverse Transcriptase (Promega) according to the manufacturer's protocol. Real-time RT-PCR was performed using SYBR green (Takara) in a QuantStudio3 Real-Time PCR System (Applied Biosystems). Expression values were normalized to the S18, GAPDH or Rpl32 mRNA level. Sequences of primers used for real-time RT-PCR are listed in Supplementary Tables 1, 2, and 3.

**Measurement of intracellular ROS**. To determine intracellular ROS, cells were treated with rotenone or O/A for 1 h after NAC pretreatment for 1 h. After incubation with 5 μM CM-H2DCFDA (Invitrogen) at 37 °C for 30 min in culture medium without FBS and recovery in complete medium for 10 min, confocal microscopy was conducted. To study mitochondria-specific ROS, cells were stained with 5 μM MitoSOX (Invitrogen) at 37 °C for 30 min. Cells were suspended in PBS-1% FBS, and then were subjected to flow cytometry by FACSVerse (BD Biosciences) using FlowJo software (TreeStar). The gating strategies for flow cytometric data analysis are shown in Supplementary Fig. 11.

**ChIP assay**. ChIP was performed according to the manufacturer's protocol (SimpleChIP® Enzymatic Chromatin IP Kit, Cell Signaling). Briefly, the chromatin/DNA protein complexes were prepared by cross-linking chromatin in *TFEB-GFP*-transfected 1.1B4 cells treated with vehicle, rotenone, or O/A for 8 h by addition of 1% formaldehyde and incubation at room temperature for 10 min. Cell extracts were then sonicated to shear DNA and immunoprecipitated using anti-GFP (1:1,000) or IgG (1:50) with constant rotation at 4 °C overnight. After the addition of 30 μL of Protein-G bead slurry to the lysates and incubation with rotation at 4 °C for 2 h, resins were washed with the ChIP buffer supplied with the kit. After elution of chromatin from the Ab/Protein-G beads with ChIP Elution Buffer and reversing cross-linking by adding 5 M NaCl and 2 μL of 20 mg/mL proteinase K, DNA was purified for PCR using primers designed to cover the putative *TFEB* CLEAR sequence (CACGTG)[50] in the human *NDP52* (-5,653 from the 5' UTR) and *OPTN* promoters (−6885 from the 5' UTR) (Supplementary Table 4). PCR consisted of 35-40 cycles of denaturation at 97 °C for 30 s, annealing at 65 °C and 55 °C for 30 s, and extension at 72 °C for 40 s for human *NDP52* and *OPTN* promoter, respectively.

**Animals**. $Tfeb^{F/F}$ mice were generated by breeding $Tfeb^{tm1a(EUCOMM)Wtsi}$ mice (Mutant Mouse Resource and Research Center) with FLPeR mice (Wellcome Trust Sanger Institute). $Tfeb^{\Delta\beta\text{-cell}}$ mice were generated by crossing $Tfeb^{F/F}$ mice with RIP-Cre mice (Jackson Laboratory). Eight-week-old male $Tfeb^{\Delta\beta\text{-cell}}$ mice were maintained in a 12-h light/12-h dark cycle and fed HFD for 12 weeks. During the observation period, mice were monitored for glucose profile and weighed. *Tfe3*-KO mice were from the Jackson Laboratory. All animal experiments were conducted in accordance with the Public Health Service Policy in the Humane Care and Use of Laboratory Animals. The mouse breeding temperature was 22 ± 2 °C, and the humidity was 50 ± 10%. Mouse experiments were approved by the IACUC of the Department of Laboratory Animal Resources of Yonsei University College of Medicine, an AAALAC-accredited unit.

**Isolation of pancreatic islets**. Primary murine islets were isolated from fasted mice using the collagenase digestion technique[2]. Briefly, after injecting 2.5 ml of collagenase P (0.8 mg/ml) into the common bile duct, the pancreas was procured and incubated in a collagenase solution at 37 °C for 13 min 20 s. After cessation of enzymatic digestion with cold HBSS-5% FBS, tissue was passed through a 400 μm sieve and then centrifuged on on 1.10, 1.085, 1.069, and 1.037 g/ml Biocoll gradients (Biochrom). Islets were collected from the interface using micropipettes.

**Intraperitoneal GTT and insulinogenic index**. Intraperitoneal GTT was performed by intraperitoneal injection of 1 g/kg of glucose after overnight fasting. Blood glucose concentrations were determined using a One Touch glucometer (Lifescan) before (0 min) and 15, 30, 60, 120, and 180 min after glucose injection. Insulinogenic index was calculated as follows: $\Delta$insulin$_{15\,min}$ (pM)/$\Delta$glucose$_{15\,min}$ (mM). ITT was conducted by injecting intraperitoneally 1 U/kg of regular insulin to fasted mice and measuring blood glucose levels at 0, 15, 30, 60, and 120 min. Serum insulin was measured using Mouse insulin ELISA kit (Shibayagi).

**Immunohistochemistry**. Tissue samples were fixed in 10% buffered formalin and embedded in paraffin. Sections of 4 μm thickness were stained for immunohistochemical analyses using anti-LC3 (MBL, 1:200), -LAMP2 (Abcam, 1:200), -TOM20 (Cell Signaling Technology, 1:200), -insulin (Invitrogen, 1:200) or 4-HNE Abs (Abcam, 1:200), and then with anti-rabbit secondary Abs conjugated to Alexa Fluor 647 (Invitrogen), Alexa Fluor 594 (Invitrogen), Alexa Fluor 568 (Invitrogen), Alexa Fluor 405 (Invitrogen) or Alexa Fluor 488 (Invitrogen). COX staining was done by incubating unfixed cryosections of the pancreas in a reaction buffer containing 10 mg of cytochrome *c*, 5 mg of diaminobenzidine tetrahydrochloride (DAB), 9 ml of sodium phosphate buffer (0.1 M, pH 7.4) and 1 ml of catalase (20 μg/ml). COX activity was visualized with a standard DAB reaction. Images were acquired with a light microscope and analyzed using Fiji[73]. To evaluate mitophagy in vivo, colocalization between LC3 and TOM20 or that between LAMP2 and TOM20 was assessed by calculating Pearson's correlation coefficient. To evaluate autophagic flux in vivo, colocalization between LC3 and LAMP2 was determined in the same way.

**Bioenergetic analysis of O₂ consumption**. O₂ consumption of isolated mouse islets was measured using a Seahorse Extracellular Flux (XFe96) Analyzer (Agilent Technologies) equipped with a spheroid microplate-compatible thermal tray according to a modification of the manufacturer's protocol. Briefly, islets were seeded into the wells of a poly-L-lysine-coated XF96 spheroid microplate (25 islets/well). Islet seeding was done by inserting a pipette tip directly into the wells of the spheroid microplate. Islets then were incubated with pre-warmed XF assay medium (Seahorse XF base DMEM supplemented with 3 mM glucose, 1% FBS, 1 mM sodium pyruvate and 2 mM glutamine) at 37 °C for 1 h in a non-CO₂ incubator. Basal respiration was measured in XF assay medium containing 3 mM glucose. Islets were then sequentially exposed to 20 mM glucose, 5 μM oligomycin, 1 μM FCCP and 5 μM rotenone/antimycin A at the indicated time points to measure glucose-stimulated, ATP-coupled, maximal and non-mitochondrial O₂ consumption, respectively. O₂ consumption was analyzed using the WaveTM software (Agilent Technologies).

**Statistical analysis**. All values are expressed as the means ± SEM of ≥ 3 independent experiments performed in triplicate. For densitometric analysis of immunoblot bands, the mean from 2 independent experiments was calculated. Two-tailed Student's *t* test was used to compare values between two groups. One-way ANOVA with Tukey's test was used to compare values between multiple groups. Two-way repeated-measures ANOVA with Bonferroni's post-hoc test was used to compare multiple repeated measurements between groups. P values < 0.05 were considered to represent statistically significant differences.

**Reporting summary**. Further information on research design is available in the Nature Research Reporting Summary linked to this article.

## Data availability

The data generated or analyzed during the current study and materials are available from the corresponding author without imposed restriction on reasonable request. Source data are provided with this paper.

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

## Acknowledgements

The authors thank Cha S-G and Park K-S for provision of probes for organelle $Ca^{2+}$ measurement, and Jae-Ho Cheong for Seahorse respirometry. ΔCnA-H151Q mutant and mRFP-LC3 were gifts from Scorrano L and Yoshimori T, respectively. *GCaMP3-*

*ML1* plasmid and ML-SI3 were provided by Xu H. INS-1 cells were from Wollheim C. *Nfe2l2*-KO mice were obtained from Yamamoto M. Flag-BirA(N-G78) and BirA(G79-C)-HA backbone vectors were from Rhee H-W. This study was supported by the National Research Foundation of Korea (NRF) grant funded by the Korea government (MSIT) (NRF-2019R1A2C3002924) and by the Bio&Medical Technology Development Program (2017M3A9G7073521). M.-S.L. is the recipient of a grant from the Faculty Research Assistance Program of Yonsei University College of Medicine (6-2016-0055) and A3 Foresight Program of the NRF (2015K2A2A6002060).

## Author contributions

M.-S.L. conceived the experiments. K.P., H.L., J.K., Y.H., Y.S.L., S.H.B., H. K., S.-W.K., and J.Y.K. conducted experiments. M.-S.L. and K.P. wrote the manuscript.

## Competing interests

M.-S.L. is the CEO of LysoTech, Inc. All authors declare no competing interests.

## Additional information

**Peer review information** *Nature Communications* thanks Hirotaka Watada and the other anonymous reviewer(s) for their contribution to the peer review this work. Peer reviewer reports are available.

