## [Peer Review File · Nature Communications]

Reviewers' Comments:

Reviewer #1:

Remarks to the Author:

Overall Comments: This paper addresses the importance of mitophagy specifically in the maintenance of β -cell function in response to metabolic stress, while the purpose and overall goal of the paper has merit, there are some concerns with the major findings of the research that need to be addressed.

Major Comments:

- It has been widely described in many tissues, including β -cells that HFD induces mitochondrial dysfunction, however here the authors describe increased mitochondrial function as shown by increases in respiration measured by Seahorse and increased COX activity. To what can the authors attribute the discrepancy in their findings with the current literature?
- It appears that about 90% of the data acquired in this manuscript are derived from semi-quantitative imaging techniques in which a high degree of subjectivity bias can occur. Images sometimes show 1-2 cells. How can the authors convey confidence in the generalizability of the data acquired?
- In Fig. 3 G, the units of measure on the 2-Y-axes are vastly different. If placed on the same scale, would the same conclusion apply?
- While the authors provide abundant evidence of reduced mitophagy in the absence of TFEB, it would be worthwhile to also demonstrate how the absence of TFEB affects autophagy. As the authors pointed out in the discussion the observed results can also be a result of autophagy or other autophagy-related processes and this cannot be discounted. Furthermore, the authors present the activation of TFEB and TFE3 following mitochondrial stress and HFD, do the authors think that despite TFE3 being activated in these conditions that it is not playing a role? Since these transcription factors have such wide and redundant roles it is important that the authors address these questions.
- The authors presented some very intriguing data that suggested Ca movement and sharing between organelles, however the conclusion that was made was that ER calcium replenishes lysosomal Ca stores during stress. The data do indicate ER to lysosomal Ca movement but the data do not support this phenomenon occurring during the actual stress, but rather after during the recovery phase to replenish lysosomal Ca and return to baseline. Note that Fig 3a, and 3g are both during the recovery phase as Rot and O/A treatment has been removed.

Minor Comments:

- Tone of the paper is at times fairly colloquial, i.e. starting sentences with "because", this should be avoided in formal writing.
- Supp fig 3b indicates TFEB and TFE3 decrease following Rot or O/A treatment, however the authors also present data that demonstrate TFEB and TFE3 activation following mitochondrial stress, how do the authors reconcile this seemingly contradictory data? Also does fig 1d support a decline in total TFEB/ TFE3 content?
- Fig 1b suggests a massive decrease in mRFP-LC3 fluorescence, consider using a different image for the vehicle condition, or providing an explanation as to why this may be. Note this decrease is not seen in the control panels of fig 2c, rather it appears to increase following Rot treatment.
- Provide clarification on fig 3h in the figure legend as it is unclear what is being quantified, blue fluorescence or red puncta? The graph does not appear to reflect the images presented, other images may be more representative.

Reviewer #2:

Remarks to the Author:

In this manuscript, Kihyoun Park et al. investigated the role of TFEB, the master regulator of lysosome biogenesis and autophagy, in mitophagy in pancreatic beta cells. Firstly, they investigated the enhanced mitophagy by mitochondrial stressors in detail. They showed that mitochondrial stressors elicit calcium efflux from lysosome to cytosol and this rise of calcium is involved in the activation of calcineurin that is known to stimulate nuclear transportation of TFEB. In this process, they demonstrated that lysosomes release calcium through their TRPML1 ion

channels by the increase of mitochondrial ROS and that ER contacts lysosomes to refill calcium ion during mitophagy. In addition, they demonstrated that uptake of extracellular calcium ion via STIM1/ORAI1 axis is essential for this process.

Secondarily, they showed the addition of palmitate to INS-1 cells raise mitochondrial ROS, TFEB activation, and mitophagy.

Finally, the authors investigated the phenotype of beta-cell-specific TFEB knock-out mice. They found that TFEB deficiency deteriorates glucose tolerance with reduced insulin secretion only under high fat diet condition. Reduced mitophagy and OCR in beta cell of TFEB knock-out mice under high fat diet suggest that decreased mitochondrial function is involved in this mechanism. Finally, they found that mitophagy receptors, NDP52 and OPTN were increased by high fat diet, but this increase was not observed in beta cells of TFEB knock-out mice suggesting that NDP52 and OPTN are downstream targets of TFEB in beta cells.

In general, the manuscript is well prepared and particularly the mechanism of calcium efflux by mitochondrial ROS production is analyzed well, following the previous reports. However, we are afraid that there are several serious concerns as described below, which requires further investigation.

Major Comments:

1. The data shown in the manuscripts do not exclude the involvement of non-selective macroautophagy regulated by TFEB and it is unclear whether disturbance of mitophagy is solely related to pancreatic beta cell failure. In general, TFEB is regarded as the regulator of non-selective macroautophagy and it is probable that mitophagy is induced as 'by-stander' of bulk autophagy. In order to stress the importance of TFEB as the regulator of mitophagy, they should also present the data showing (1) that other organelles except for mitochondria are not degraded and (2) how non-selective macroautophagy is activated, following ROS-induced TFEB activation. If those data show involvement of TFEB-mediated macroautophagy, the authors cannot exclude the possibility that mitophagy is promiscuously induced along with macroautophagy, making it difficult to conclude that selective mitophagy is regulated by TFEB. In addition, it is unclear that the TFEB-induced mitophagy is independent of PINK1-Parkin system. The authors suggested that transcription of NDP52 and OPTN is regulated by TFEB, however, it should be noted that the roles of those receptors have been described primarily in PINK1-Parkin dependent mitophagy (Nature 524, 309–314, 2015).

2. It is uncertain that palmitate (PA) acts as an inducer of ROS and ROS-mediated mitophagy, which is primarily demonstrated by administering mitochondrial stressors such as rotenone in Figures 1-3. For example, treatment of PA on INS-1 increased MitoSox fluorescence reflecting ROS accumulation only after 24-hour treatment (Fig 4a), whereas TFEB translocation was confirmed even in 4 hours, when ROS accumulation was not demonstrated yet (Fig 4d). PA has several bioactive effects such as ligand of G-protein coupled receptor, source of various fatty acids, occurring signal transduction, and metabolic fuels. If the authors consider PA as an alternative inducer of mitophagy to mitochondrial stressors, they should also evaluate those pathways considered as PA-regulated and show that mitophagy is not disturbed by them.

3. The authors evaluated the effect of extracellular calcium in TFEB-induced mitophagy by adding Ca chelators in extracellular media or inhibiting ORAI. TFEB-induced mitophagy was also significantly inhibited following those procedures and the extent of inhibition is comparable to or even larger than chelating intracellular calcium or inhibiting release of lysosomal calcium (compare Supplementary figure 2a and 5b). These data might suggest that it is extracellular calcium that matters for TFEB-induced mitophagy, not lysosomal one. To clarify if extracellular calcium is essential for TFEB-induced mitophagy, the authors should examine whether increase of lysosomal calcium content followed by decrease of one in ER after administration of the mitochondrial stressors is inhibited by extracellular Ca chelators.

Minor Comments:

1. References No. 24: The name of the journal "Developmental Cell" is missing.
2. References No. 41: The year of publication is 2013, not 1985.

Reviewer #3:

Remarks to the Author:

The manuscript by Kihyoun Park et al. details Regulation and role of TFEB in mitophagy and the function of pancreatic beta-cells. The manuscript finds that beta-cell TFEB undergoes nuclear translocation following inhibition of ATP synthase or complex I. The data also finds that calcium is required for TFEB translocation and that lysosomes are a potential calcium source for TFEB translocation. TFEB is also shown to regulate mitochondrial function in a beta-cell-specific Tfeb knockout model. The manuscript includes a significant amount of data related to how activation of TFEB regulates beta-cell mitochondrial function. Although many of the findings regarding the calcium-dependence of TFEB nuclear translocation have previously been described in other cell types, the beta-cell specific KO of TFEB provides important insights into the roles of TFEB in beta-cell mitochondrial function and insulin secretion. There are some issues that require attention, which are specified below.

Major:

The results section mentions that, "Immunoblotting showed that INS-1 cells express Parkin (Supplementary Fig. 1a), which is necessary for mitophagy after treatment with mitochondrial stressors." However, this is not necessarily the entire story as Parkin knockout in primary mouse beta-cells does not impact mitophagy (see PMID: 30877201).

Pancreatic beta-cell organelle Ca²⁺ levels are maintained by either calcium pumps or organelle mediated transfer of calcium down its concentration gradient. Therefore, inhibiting ATP production would be predicted to reduce ER calcium by inhibiting SERCA and thus lysosomal calcium. Indeed, thapsigargin a SERCA inhibitor has been shown to cause TFEB nuclear translocation (PMID: 25720963). If STIM-1 interacts with Orai1 during ATP synthase inhibition or complex I inhibition, then this should increase cytoplasmic calcium because SERCA cannot be activated in the absence of ATP and store operated calcium influx is still active. It is thus unclear how TRPML1 could be the reason for enhanced cytoplasmic calcium elevation following ATP synthase inhibition or complex I inhibition. SERCA should be limited in terms of activity during ATP synthase inhibition or complex I inhibition and thus transfer of calcium between ER and lysosome should also be low. The authors need to confirm that the mechanism of increased cytoplasmic calcium and TFEB nuclear translocation is not just simply due to ER calcium depletion.

The manuscript uses a single wavelength dye (Fluo-3-AM) to monitor relative changes in cytosolic calcium (such as that shown in supplemental figure 2c). Dye loading of cells and/or leakage from cells may be impacted by the conditions used to monitor intracellular calcium. Therefore, to make any conclusive comments about intracellular calcium a ratiometric calcium indicator is required. Furthermore, determining total ER calcium with CEPIA1er (a single wavelength calcium indicator) is not great. Not only can this indicator bleach but it also shows expression differences in different settings. Therefore, in order to make claims about cytoplasmic or ER calcium levels, confirmation with ratiometric indicators is required.

The RIP-cre line has been shown to have some problems such as glucose intolerance as well as expression of an Hgh minigene from the RIP-cre transgene (PMID: 16326700). The controls used for all of the RIP-cre containing mice was the Tfeb^{fl/fl}. One issue with these findings is that they do not use the RIP-cre as a control for the experiments and thus may be representing beta-cell perturbations caused by the RIP-cre transgene instead of beta-cell specific knockout of Tfeb.

The rolling circle amplification indicates that the association of ORP1L and VAPA may increase following ATP synthase inhibition or complex I inhibition. However, this is not conclusive confirmation that lysosomes and ER show greater contact following ATP synthase inhibition or complex I inhibition, which should be substantiated with organelle contacts utilizing another technique.

Xestospongins are not truly selective for IP3Rs and has been shown to inhibit SERCA (PMID: 10892566). Therefore, the claim that ER calcium refills lysosomes cannot be made by adding 20 micromolar Xestospongins. Moreover, Dantrolene also has off target impacts such as activation of the mitochondrial carnitine/acylcarnitine carrier EC50 9.3 micromolar (PMID: 31775359) and the current manuscript uses 20 micromolar dantrolene. Therefore, due to the off-target influences of these molecules, further confirmation of the link between ER calcium and lysosomal calcium is required.

TPEN can impact zinc levels, oxidative stress, and mitochondrial function (PMID: 12724524). Therefore, the claim that TPEN treatment "supports the role of ER Ca²⁺ in these processes," is not substantiated with this treatment.

It is not clear how colocalization with the confocal images was determined. The manuscript needs to detail the analysis and statistical method used to confirm colocalization.

Minor:

On page 7 TRPML1 is spelled TRMPM1 on the second sentence from the bottom of the page.

Figure 4d shows a merge image at 4 hours post PA treatment however the only channel shown is green and there are no nuclei stained with dapi.

As Oregon Green 488 BAPTA-1 Dextran is sensitive to pH, the authors need to ensure that the changes of [Ca²⁺]_{Lys} with this technique is not due to changes in pH.

REVIEWER COMMENTS

Reviewer #1 (Remarks to the Author):

Overall Comments: This paper addresses the importance of mitophagy specifically in the maintenance of β -cell function in response to metabolic stress, while the purpose and overall goal of the paper has merit, there are some concerns with the major findings of the research that need to be addressed.

Major Comments:

- It has been widely described in many tissues, including β -cells that HFD induces mitochondrial dysfunction, however here the authors describe increased mitochondrial function as shown by increases in respiration measured by Seahorse and increased COX activity. To what can the authors attribute the discrepancy in their findings with the current literature?

Ans) We agree with the reviewer's comment that HFD induces mitochondrial stress such as mitochondrial ROS production in several tissues including islet tissues. However, previous papers reported that mitochondrial O₂ consumption and expression of mitochondrial complex proteins including COX were increased but not decreased in the heart, pancreatic islets or lymphocytes of HFD-fed animals (Rocha C et al., J Physiol Biochem 75:65, 2019; Roat R et al. PLOS One 9:e86815, 2014; Liu F et al., Diabetes Metab Syndr Obes 13:2701, 2020; Yu H et al. Eur J Nutr 57:1957, 2018; Moreno-Fernandez ME et al., Cell Metab, 33:1187, 2021), which could be due to fatty acid-induced uncoupling, O₂ shunting to ROS pathway, or adaptation to mitochondrial stress (Rocha C et al., J Physiol Biochem 75:65, 2019). Increased mitophagic activity in cardiac tissue of HFD-fed mice has also been reported (Tong M et al. Circ Res 125:1360, 2019). Thus, increased mitochondrial O₂ consumption, increased mitochondrial complex proteins and increased mitophagy observed in tissues in vivo, which are in line with our results in this paper, are adaptive responses and not incompatible with

mitochondrial stress by HFD feeding. While this part has been discussed in our previous submission, we further extended our discussion in lines 9-15 of page 18 and added more references (new references 43, 53, 54 & 55).

- It appears that about 90% of the data acquired in this manuscript are derived from semi-quantitative imaging techniques in which a high degree of subjectivity bias can occur. Images sometimes show 1-2 cells. How can the authors convey confidence in the generalizability of the data acquired?

Ans) We agree with the reviewer's comment that determination of Ca^{2+} content was mostly semiquantitative. Thus, we determined Ca^{2+} fluorescence again employing a ratiometric assay using Fura-2, and the F340/F380 ratios were converted to Ca^{2+} concentrations using a published protocol (Grynkiewicz, G et al. J Biol Chem 260:3440, 1985). Two sets of data showed similar tendency. All Ca^{2+} data determined using Fluo-3 were replaced by those determined using Fura-2 (new Fig. 2g, 2h, 3b, 4c, Supplementary Fig. 3b, and Supplementary Fig. 5b,d).

- In Fig. 3 G, the units of measure on the 2-Y-axes are vastly different. If placed on the same scale, would the same conclusion apply?

Ans) Labels of the Y axis represent relative fluorescence of ER Ca^{2+} fluorescence (GEM-CEPIA1er) and lysosomal Ca^{2+} fluorescence (Oregon G BAPTA-1 Dextran) (Fig. 3e in the revised manuscript). Since they show relative scales based on the respective control fluorescence, their absolute values of fluorescence cannot be compared directly. Even if they are placed on the same scale, the tendency is the same. However, if they are placed on the same scale, it would be difficult to observe the tendency and slope of the fluorescence curves or relationship between the curves because their absolute values are very different from each other, as was shown below (see figures in the next page).

- While the authors provide abundant evidence of reduced mitophagy in the absence of TFEB, it would be worthwhile to also demonstrate how the absence of TFEB affects autophagy. As the authors pointed out in the discussion the observed results can also be a result of autophagy or other autophagy-related processes and this cannot be discounted. Furthermore, the authors present the activation of TFEB and TFE3 following mitochondrial stress and HFD, do the authors think that despite TFE3 being activated in these conditions that it is not playing a role? Since these transcription factors have such wide and redundant roles it is important that the authors address these questions.

Ans) We agree with the reviewer’s comment that autophagy or other selective autophagy might be involved in the TFEB-mediated protection of β -cell function. Thus, we determined autophagic flux in pancreatic islets of each group of mice by studying colocalization of LC3 with LAMP2. However, flux of autophagy was not increased in pancreatic islets by HFD feeding in vivo. These results are consistent with a previous paper showing increase of mitophagic activity but not autophagic activity after HFD feeding for more than 2 months (Tong M et al. *Circ Res* 125:1360, 2019). We also determined autophagic activity and ER-phagic activity in vitro using *mRFP-GFP-LC3* and *TetOn-mCherry- GFP-RAMP4* plasmid, respectively. Autophagic activity or ER-phagic activity in vitro was not increased by mitochondrial stressor or PA treatment, which is in agreement with previous papers (Hernández-Cáceres MP et al. *Mol. Cell. Oncol.* 25, 1789418, 2020; Lim Y et al., *Mol. Brain* 14, 65, 2021). These results could be due to the effects of altered intracellular environment by

organelle stressors such as impaired autophagosome-lysosome fusion by rotenone or PA (Koga H et al. *FASEB J.* 24, 3052, 2010; Wu F et al. *Neuroscience* 284, 900, 2015), which could modify TFEB-dependent or -independent autophagic activity. Induction of mitophagic receptors OPTN, NDP52 such as by HFD feeding (Fig. 6c) might be involved in the preferential induction of mitophagy by HFD, compared to bulk autophagy or ER-phagy. These results were incorporated as new Supplementary Fig. 2c,d and Supplementary Fig. 9c, with discussion in lines 2-8 from the bottom of page 12, lines 4-8 of page 14 and lines 1-8 of page 18.

We also agree with the reviewer's suggestion that TFE3 might exert non-redundant effects in addition to TFEB in the response of β -cells to metabolic stress. Indeed, we have been crossing β -cell-specific *Tfeb*-KO mice (*Tfeb* ^{$\Delta\beta$ -cell} mice) to *Tfe3*-KO mice. While *Tfeb*^{F/F}/*Tfe3*-KO and *Tfeb* ^{$\Delta\beta$ -cell}/*Tfe3*-KO mice appeared to have slightly impaired glucose tolerance (AUC of GTT curves) compared to *Tfeb*^{F/F}/*Tfe3* wild-type (WT) mice and *Tfeb* ^{$\Delta\beta$ -cell}/*Tfe3* WT, respectively, the differences were statistically insignificant in both normal chow- and HFD-fed conditions. These results suggesting no significant additional independent role of endogenous TFE3 in metabolic profile of *Tfeb* ^{$\Delta\beta$ -cell} mice were incorporated as new Supplemental Fig. 10 with discussion (lines 1-3 from the bottom of page 18 & lines 1-3 of page 19).

- The authors presented some very intriguing data that suggested Ca movement and sharing between organelles, however the conclusion that was made was that ER calcium replenishes lysosomal Ca stores during stress. The data do indicate ER to lysosomal Ca movement but the data do not support this phenomenon occurring during the actual stress, but rather after during the recovery phase to replenish lysosomal Ca and return to baseline. Note that Fig 3a, and 3g are both during the recovery phase as Rot and O/A treatment has been removed.

Ans) As suggested, we determined lysosomal Ca²⁺ ([Ca²⁺]_{Lys}) during the actual stress, i.e., in the continued presence of mitochondrial stressors. In the presence of continuous mitochondrial stressors, recovery of [Ca²⁺]_{Lys} could not be observed probably because of ongoing release of lysosomal Ca²⁺. In contrast, decline of ER Ca²⁺ content was still observed. These results were incorporated as new Supplementary Fig. 6d.

Of course, this result does not indicate that recovery of [Ca²⁺]_{Lys} or ER→lysosome Ca²⁺

refilling does not occur. Such changes just could not be observed due to ongoing release of lysosomal Ca^{2+} . Thus, to observe such changes, further release of lysosomal Ca^{2+} should be stopped or lysosomal Ca^{2+} level should be “clamped”.

Minor Comments:

- Tone of the paper is at times fairly colloquial, i.e. starting sentences with “because”, this should be avoided in formal writing.

Ans) As suggested, we changed the “because” at the starting point sentence to other words, or changed the sentence to another way.

- Supp fig 3b indicates TFEB and TFE3 decrease following Rot or O/A treatment, however the authors also present data that demonstrate TFEB and TFE3 activation following mitochondrial stress, how do the authors reconcile this seemingly contradictory data? Also does fig 1d support a decline in total TFEB/ TFE3 content?

Ans) We agree with the reviewer’s comment that decline of total TFEB/TFE3 after O/A treatment is not entirely compatible with nuclear translocation of TFEB/TFE3. However, as was shown in new Supplementary Fig. 4b, still some total and dephosphorylated TFEB/TFE3 can be seen, suggesting degradation of TFEB/TFE3 by O/A is not 100%. Some remaining dephosphorylated TFEB/TFE3 could explain nuclear TFEB/TFE3. In Fig. 1d, fluorescence of nuclear TFEB and remaining cytosolic TFEB after O/A treatment appears to be weaker than that of cytosolic TFEB before O/A treatment (after vehicle treatment), while we don't have quantitative data of Fig. 1d because quantitative comparison between cytosolic TFEB and nuclear TFEB staining is not theoretically feasible.

- Fig 1b suggests a massive decrease in mRFP-LC3 fluorescence, consider using a different image for the vehicle condition, or providing an explanation as to why this may be. Note this decrease is not seen in the control panels of fig 2c, rather it appears to increase following Rot treatment.

Ans) As the reviewer indicated, we observed a large RFP-LC3 staining in the nuclei in both Fig. 1b and Fig. 2c. We don’t understand the identity of this structure. However, we have

observed the same structure in the nuclei in our previous experiment (Kim J et al. Nat Commun 12:183, 2021, Fig. 1a). In the previous papers by others also, such structure has been observed in RFP-LC3-stained HEK293 cells without any treatment (Deng L et al., Cell Death Dis 4:e614, 2013). In our experience, appearance of such structure is dependent on the cell types transfected. Such nuclear structure seems to disappear after rotenone or O/A treatment as was observed in the current manuscript but not after starvation or chemical autophagy enhancer treatment (Kim J et al. Nat Commun 12:183, 2021, Fig. 1a). While we don't have idea regarding the mechanism of formation of such nuclear structures, we speculate that some nuclear proteins or structures undergoing constitutive autophagy or changes of pH in the nuclei might be responsible for such changes.

- Provide clarification on fig 3h in the figure legend as it is unclear what is being quantified, blue fluorescence or red puncta? The graph does not appear to reflect the images presented, other images may be more representative.

Ans) We clarified what the red puncta represents in the legend of Fig. 3h (Fig. 3f in the revised figures), as suggested (e.g. red bar after Rot treatment and blue bar after O/A treatment). We also added that the blue represents DAPI and red represents VAPA-ORPL1 puncta below Fig. 3f. While the intensity of red dots 4 h after O/A treatment is weaker than those in other fields, the number of red dots 4 h after O/A treatment was higher than those in other fields, which is the reason the graph does not appear to reflect the images. We replaced the figure with a new one showing red dots with more intense fluorescence. Fig. 3f after treatment with vehicle only was also changed to represent the experimental results better.

Reviewer #2 (Remarks to the Author):

In this manuscript, Kihyoun Park et al. investigated the role of TFEB, the master regulator of lysosome biogenesis and autophagy, in mitophagy in pancreatic beta cells. Firstly, they investigated the enhanced mitophagy by mitochondrial stressors in detail. They showed that mitochondrial stressors elicit calcium efflux from lysosome to cytosol and this rise of calcium is involved in the activation of calcineurin that is known to stimulate nuclear transportation of TFEB. In this process, they demonstrated that lysosomes release calcium through their TRPML1 ion channels by the increase of mitochondrial ROS and that ER contacts lysosomes to refill calcium ion during mitophagy. In addition, they demonstrated that uptake of

extracellular calcium ion via STIM1/ORAI1 axis is essential for this process.

Secondarily, they showed the addition of palmitate to INS-1 cells raise mitochondrial ROS, TFEB activation, and mitophagy.

Finally, the authors investigated the phenotype of beta-cell-specific TFEB knock-out mice. They found that TFEB deficiency deteriorates glucose tolerance with reduced insulin secretion only under high fat diet condition. Reduced mitophagy and OCR in beta cell of TFEB knock-out mice under high fat diet suggest that decreased mitochondrial function is involved in this mechanism. Finally, they found that mitophagy receptors, NDP52 and OPTN were increased by high fat diet, but this increase was not observed in beta cells of TFEB knock-out mice suggesting that NDP52 and OPTN are downstream targets of TFEB in beta cells.

In general, the manuscript is well prepared and particularly the mechanism of calcium efflux by mitochondrial ROS production is analyzed well, following the previous reports. However, we are afraid that there are several serious concerns as described below, which requires further investigation.

Major Comments:

1. The data shown in the manuscripts do not exclude the involvement of non-selective macroautophagy regulated by TFEB and it is unclear whether disturbance of mitophagy is solely related to pancreatic beta cell failure. In general, TFEB is regarded as the regulator of non-selective macroautophagy and it is probable that mitophagy is induced as 'by-stander' of bulk autophagy. In order to stress the importance of TFEB as the regulator of mitophagy, they should also present the data showing (1) that other organelles except for mitochondria are not degraded and (2) how non-selective macroautophagy is activated, following ROS-induced TFEB activation. If those data show involvement of TFEB-mediated macroautophagy, the authors cannot exclude the possibility that mitophagy is promiscuously induced along with macroautophagy, making it difficult to conclude that selective mitophagy is regulated by TFEB. In addition, it is unclear that the TFEB-induced mitophagy is independent of PINK1-Parkin system. The authors suggested that transcription of NDP52 and OPTN is regulated by TFEB, however, it should be noted that the roles of those receptors have been described primarily in PINK1-Parkin dependent mitophagy (Nature 524, 309–314, 2015).

Ans) We agree with the reviewer's comments that bulk autophagy or other types of selective autophagy could be activated by TFEB and affect the β -cells or systemic metabolism. Thus,

we additionally studied whether autophagic flux or ER-phagic flux is enhanced by treatment with mitochondrial stressors or PA, as suggested. We observed that autophagic activity determined by red puncta counting after *mRFP-GFP-LC3* transfection was not increased by PA (new Supplementary Fig. 2c), which is in agreement with previous papers (Hernández-Cáceres MP et al. *Mol. Cell. Oncol.* 25, 1789418, 2020; Koga H et al. *FASEB J.* 24, 3052, 2010). ER-phagy flux determined by red puncta counting after TetOn-mCherry-GFP-RAMP4 transfection was also not increased by rotenone, O/A or PA (new Supplementary Fig. 2d), consistent with a previous paper (Lim Y et al., *Mol. Brain* 14, 65, 2021). Furthermore, in vivo autophagic activity determined by colocalization of LAMP2 with LC3 was not increased in pancreatic islets of HFD-fed mice (new Supplementary Fig. 9c), while mitophagic activity was markedly increased (previous Fig. 5b). These results are in agreement with a previous paper (Tong et al. *Cir Res* 124:1360 2019) reporting increased mitophagic activity but no increased bulk autophagic activity in cardiac tissue after 2 months of HFD feeding and support that β -cell protective effect of TFEB in HFD-fed mice was mostly due to due to mitophagy induction. Mechanism of selective activation of mitophagy by mitochondrial stressors despite activation of TFEB is not clear, and suggests that additional factors or changes of intracellular milieu after mitochondrial stressor or PA treatment might affect flux of global or selective autophagy (ER-phagy or mitophagy) in a differential manner. Induction of NDP52 or optineurin by mitochondrial stressors or PA might be related to preferential induction of mitophagy compared to autophagy or ER-phagy. These results were incorporated as new Supplementary Fig. 2c,d and Supplementary Fig. 9c, with discussion in lines 2-8 from the bottom of page 12, lines 4-8 of page 14 and lines 1-8 of page 18.

We are not claiming that TFEB-mediated mitophagy is Parkin-independent. Since a previous paper reported no change of pancreatic β -cell function in mice with β -cell-specific *Parkin* KO (Corsa CAS et al. *J Biol Chem* 294:7296, 2019), Parkin-independent mitophagy might occur and could explain the no change of β -cell function by *Parkin* KO. However, TFEB-dependent mitophagy in the current investigation would most likely be Parkin-dependent mitophagy, since TFEB nuclear translocation does not occur in cells without Parkin expression.

2. It is uncertain that palmitate (PA) acts as an inducer of ROS and ROS-mediated mitophagy, which is primarily demonstrated by administering mitochondrial stressors such as rotenone

in Figures 1-3. For example, treatment of PA on INS-1 increased MitoSox fluorescence reflecting ROS accumulation only after 24-hour treatment (Fig 4a), whereas TFEB translocation was confirmed even in 4 hours, when ROS accumulation was not demonstrated yet (Fig 4d). PA has several bioactive effects such as ligand of G-protein coupled receptor, source of various fatty acids, occurring signal transduction, and metabolic fuels. If the authors consider PA as an alternative inducer of mitophagy to mitochondrial stressors, they should also evaluate those pathways considered as PA-regulated and show that mitophagy is not disturbed by them.

Ans) We agree with the reviewer's comment that mitochondrial ROS and TFEB activation showed different time course in our previous submission. Then, in Fig. 4a of our previous submission, there was an apparent increase of mitochondrial ROS in the scattergram (please see the left panel of previous Fig. 4a) but no significant increase of mitochondrial ROS in the bar graph because of an outlier value in the control experiment (right of previous Fig. 4a). Hence, the previous bar graph was not representative of real ROS measured. We carefully conducted the experiment again, and now the time lapse of ROS and Tfeb activation is coherent in that significant ROS accumulation was observed after 4 h of treatment with PA in both scattergram and bar graph, which was incorporated as new Fig. 4a. In addition, to prove the causality between mitochondrial ROS accumulation and TFEB activation by PA, we studied the effect of NAC and MitoTEMPO on TFEB activation after treatment with PA for 4 h. TFEB activation after treatment with PA was suppressed by NAC or MitoTEMPO, supporting the role of mitochondrial ROS in TFEB activation. These results were incorporated as new Fig. 4f.

We agree with the reviewer's comments that PA can affect other signaling pathway such as fatty acid receptor/GPCR or can be converted to other lipid intermediates through palmitoyl-CoA or sphingosine/ceramide pathway. We thus studied the effect of DC260126 (GPCR antagonist), Triacsin C (acyl-CoA synthetase inhibitor), Fumonisin B1 (inhibitor of sphingosine *N*-acyltransferase) or Myriocin (serine palmitoyltransferase inhibitor) (Han MS et al. *J Lipid Res* 49:87, 2008). These reagents did not reduce mitochondrial ROS accumulation or TFEB nuclear translocation by PA, suggesting that mitochondrial stress by PA itself is the most likely event leading to mitophagy. These results were incorporated as new Supplementary Fig. 8 and discussed in lines 1-9 of page 13.

3. The authors evaluated the effect of extracellular calcium in TFEB-induced mitophagy by adding Ca chelators in extracellular media or inhibiting ORAI. TFEB-induced mitophagy was also significantly inhibited following those procedures and the extent of inhibition is comparable to or even larger than chelating intracellular calcium or inhibiting release of lysosomal calcium (compare Supplementary figure 2a and 5b). These data might suggest that it is extracellular calcium that matters for TFEB-induced mitophagy, not lysosomal one. To clarify if extracellular calcium is essential for TFEB-induced mitophagy, the authors should examine whether increase of lysosomal calcium content followed by decrease of one in ER after administration of the mitochondrial stressors is inhibited by extracellular Ca chelators.

Ans) As suggested, we measured lysosomal Ca^{2+} recovery without extracellular Ca^{2+} (by incubating in KRB or in the presence of EGTA). Lysosomal Ca^{2+} recovery was significantly attenuated without extracellular Ca^{2+} , suggesting that the source of lysosomal Ca^{2+} recovery is extracellular Ca^{2+} , as suggested. These results were incorporated as new Supplementary Fig. 6a.

However, these results do not suggest that lysosomal Ca^{2+} does not matter for TFEB-induced mitophagy. Indeed, the decrease of ER Ca^{2+} content which was observed 1 h after rotenone or O/A treatment without extracellular Ca^{2+} (new Fig. 3d) was not observed in the presence of extracellular Ca^{2+} (new Supplementary Fig. 6c), which strongly support that extracellular Ca^{2+} enters to ER through CRAC activation during mitochondrial stress. These data suggest that extracellular Ca^{2+} influx goes to ER, the largest Ca^{2+} reservoir unless SERCA is inhibited (Suzuki J et al. Nat Commun 5:4153, 2014). If SERCA activity of ER is inhibited, extracellular Ca^{2+} influx into ER is impaired (Suzuki J et al. Nat Commun 5:4153, 2014). Since extracellular Ca^{2+} went to ER well during mitochondrial stress as stated above, SERCA was probably not inhibited by mitochondrial stressors. And extracellular Ca^{2+} went to ER rather than directly to cytoplasm. Thus, while extracellular Ca^{2+} influx matters for TFEB-induced mitophagy as suggested by the reviewer, extracellular Ca^{2+} \rightarrow CRAC \rightarrow ER \rightarrow lysosome pathway seems to contribute to the mitochondrial stress-induced mitophagy. These results were incorporated as new Supplementary Fig. 6b,c and new Fig. 3d with more discussion in lines 3-10 of page 10.

While we described ER Ca^{2+} data in our previous Fig. 3f as “90 min after rotenone or O/A treatment without extracellular Ca^{2+} ”, we found that the real incubation time varied between 60-90 min and, in fact, more close to 60 min. Thus, we changed the legend for the previous

Fig. 3f (new Supplementary Fig. 6b) as “incubation for 1 h”.

Minor Comments:

1. References No. 24: The name of the journal “Developmental Cell” is missing.
2. References No. 41: The year of publication is 2013, not 1985.

Ans) We would like to thank the reviewer for such a kind and thorough checking. We are deeply sorry for our mistakes.

Reviewer #3 (Remarks to the Author):

The manuscript by Kihyoun Park et al. details Regulation and role of TFEB in mitophagy and the function of pancreatic beta-cells. The manuscript finds that beta-cell TFEB undergoes nuclear translocation following inhibition of ATP synthase or complex I. The data also finds that calcium is required for TFEB translocation and that lysosomes are a potential calcium source for TFEB translocation. TFEB is also shown to regulate mitochondrial function in a beta-cell-specific Tfeb knockout model. The manuscript includes a significant amount of data related to how activation of TFEB regulates beta-cell mitochondrial function. Although many of the findings regarding the calcium-dependence of TFEB nuclear translocation have previously been described in other cell types, the beta-cell specific KO of TFEB provides important insights into the roles of TFEB in beta-cell mitochondrial function and insulin secretion. There are some issues that require attention, which are specified below.

Major:

The results section mentions that, “Immunoblotting showed that INS-1 cells express Parkin (Supplementary Fig. 1a), which is necessary for mitophagy after treatment with mitochondrial stressors.” However, this is not necessarily the entire story as Parkin knockout in primary mouse beta-cells does not impact mitophagy (see PMID: 30877201).

Ans) We agree with the reviewer's comment that β -cell-specific KO of Parkin did not affect mitophagy or metabolic profile. In fact, we observed similar features in that β -cell-specific PINK1 did not affect metabolic profile after HFD feeding (Park K et al. unpublished data). We presume that adaptive changes in β -cell specific Parkin-KO or PINK1-KO might occur and affect mitophagy or metabolic profile, which has been discussed in the lines 3-7 of page 19 of the revised manuscript.

Pancreatic beta-cell organelle Ca^{2+} levels are maintained by either calcium pumps or organelle mediated transfer of calcium down its concentration gradient. Therefore, inhibiting ATP production would be predicted to reduce ER calcium by inhibiting SERCA and thus lysosomal calcium. Indeed, thapsigargin a SERCA inhibitor has been shown to cause TFEB nuclear translocation (PMID: 25720963). If STIM-1 interacts with Orai1 during ATP synthase inhibition or complex I inhibition, then this should increase cytoplasmic calcium because SERCA cannot be activated in the absence of ATP and store operated calcium influx is still active. It is thus unclear how TRPML1 could be the reason for enhanced cytoplasmic calcium elevation following ATP synthase inhibition or complex I inhibition. SERCA should be limited in terms of activity during ATP synthase inhibition or complex I inhibition and thus transfer of calcium between ER and lysosome should also be low. The authors need to confirm that the mechanism of increased cytoplasmic calcium and TFEB nuclear translocation is not just simply due to ER calcium depletion.

Ans) If the decrease of ATP by mitochondrial stressors inhibits SERCA and thereby increase cytosolic Ca^{2+} content, then ER Ca^{2+} content should be decreased even in the presence of extracellular Ca^{2+} (Suzuki J et al. Nat Commun 5:4153, 2014). Thus, we conducted a new experiment to determine $[\text{Ca}^{2+}]_{\text{ER}}$ in the presence of extracellular Ca^{2+} . ER Ca^{2+} content after treatment with rotenone or O/A in the presence of extracellular Ca^{2+} did not decrease, while cytosolic Ca^{2+} content was increased and lysosomal Ca^{2+} content was decreased 1 h after treatment with rotenone or O/A (see new Fig. 2f-h). In contrast, a significant decrease of ER Ca^{2+} 1 h after treatment with mitochondrial stressors was well observed in the absence of extracellular Ca^{2+} (Fig. 3d in the revised manuscript). Thus, these results suggest that ER Ca^{2+} emptying by mitochondrial stressors is due to the absence of CRAC and is not related to SERCA inhibition by rotenone or O/A. These results were incorporated as new Supplementary Fig. 6c with discussion (lines 3-10 of page 10).

The manuscript uses a single wavelength dye (Fluo-3-AM) to monitor relative changes in cytosolic calcium (such as that shown in supplemental figure 2c). Dye loading of cells and/or leakage from cells may be impacted by the conditions used to monitor intracellular calcium. Therefore, to make any conclusive comments about intracellular calcium a ratiometric

calcium indicator is required. Furthermore, determining total ER calcium with CEPIA1er (a single wavelength calcium indicator) is not great. Not only can this indicator bleach but it also shows expression differences in different settings. Therefore, in order to make claims about cytoplasmic or ER calcium levels, confirmation with radiometric indicators is required.

Ans) We agree with the reviewer's suggestion that ratiometric measurement of Ca^{2+} content is necessary. As suggested, we additionally conducted ratiometric measurement of cytosolic Ca^{2+} content using Fura-2, which was incorporated in Fig. 2g, 2h, 3b, 4c, Supplementary Fig. 3b, and Supplementary Fig. 5b,d. Indeed, all cytosolic Ca^{2+} data using Fluo-3 were replaced with that using Fura-2. We also measured ER Ca^{2+} content using D1ER, a FRET-based probe allowing ratiometric measurement of ER Ca^{2+} content. These results were incorporated in Fig. 3d and Supplementary Fig. 6c.

The RIP-cre line has been shown to have some problems such as glucose intolerance as well as expression of an Hgh minigene from the RIP-cre transgene (PMID: 16326700). The controls used for all of the RIP-cre containing mice was the Tfebfl/fl. One issue with these findings is that they do not use the RIP-cre as a control for the experiments and thus may be representing beta-cell perturbations caused by the RIP-cre transgene instead of beta-cell specific knockout of Tfeb.

Ans) We agree with the reviewer's concern that RIP-Cre might cause some problems associated with growth hormone minigene expression. However, we have been using RIP-Cre mice for a long time, and we have never observed significant differences in blood glucose, insulin level or glucose tolerance between RIP-Cre+ and RIP-Cre- littermates (Quan W et al., Diabetologia 55:392, 2012, ESM Fig. 4; Kim S et al. PNAS 104:1913, 2007, Fig. S9). Thus, we believe that transgene effect of RIP-Cre did not affect the results in the current investigation.

The rolling circle amplification indicates that the association of ORP1L and VAPA may increase following ATP synthase inhibition or complex I inhibition. However, this is not conclusive confirmation that lysosomes and ER show greater contact following ATP synthase inhibition or complex I inhibition, which should be substantiated with organelle contacts utilizing another technique.

Ans) To provide another evidence of contact between lysosomal protein and ER protein, we conducted a new experiment employing a modification of Contact-ID technology using a split-pair system of pBir biotin ligase (Kwak C et al. PNAS 117:12109, 2020).

Briefly, we inserted DNA fragments encoding VAPA and STARD3 into Age1-XhoI site of Flag-BirA(N-G78) and Hind3-BamHI site of BirA(G79-C)-HA vectors, respectively. After transfection of INS-1 cells with the resultant two plasmids, cells were incubated with biotin, and then treated with mitochondrial stressors. Confocal microscopy after FITC-streptavidin incubation demonstrated formation of biotin conjugate after rotenone or O/A treatment of cells transfected with Flag-BirA(N-G78)-VAPA and STARD3-BirA(G79-C)-HA, suggesting contact between VAPA, an ER protein, and STARD3, a lysosomal protein. These result were incorporated as new Supplementary Fig. 6e and the method of modified contact-ID was described in page 24-25 in detail.

Xestospongine C is not truly selective for IP3Rs and has been shown to inhibit SERCA (PMID: 10892566). Therefore, the claim that ER calcium refills lysosomes cannot be made by adding 20 micromolar Xestospongine C. Moreover, Dantrolene also has off target impacts such as activation of the mitochondrial carnitine/acylcarnitine carrier EC50 9.3 micromolar (PMID: 31775359) and the current manuscript uses 20 micromolar dantrolene. Therefore, due to the off-target influences of these molecules, further confirmation of the link between ER calcium and lysosomal calcium is required.

TPEN can impact zinc levels, oxidative stress, and mitochondrial function (PMID: 12724524). Therefore, the claim that TPEN treatment “supports the role of ER Ca²⁺ in these processes,” is not substantiated with this treatment.

Ans) We agree with the reviewer’s suggestion that reduced lysosomal Ca²⁺ recovery after treatment with Xestospongine C, dantrolene or TPEN does not confirm the role of ER Ca²⁺ in this process. To further corroborate the role of ER Ca²⁺ in the recovery of lysosomal Ca²⁺ after removal of mitochondrial stressors, we treated cells with Thapsigargin emptying ER Ca²⁺ store (Garrity AG et al. eLife 5:15887, 2016). After pretreatment with thapsigargin, recovery of lysosomal Ca²⁺ content after removal of mitochondrial stressors was abrogated, supporting an important role of ER Ca²⁺ in the recovery of lysosomal Ca²⁺. These results were incorporated as Supplementary Fig. 6a with discussion (lines 3-10 from the bottom of

page 9).

It is not clear how colocalization with the confocal images was determined. The manuscript needs to detail the analysis and statistical method used to confirm colocalization.

Ans) We agree with the reviewer's comment that statistical analysis is required to present objective data of colocalization. We calculated Pearson's correlation coefficient as a measure of colocalization, which was incorporated in Fig. 5a,b, Supplemental 7a and Supplemental Fig. 9c. We also added a description of in vitro method of determining colocalization in lines 4-5 from the bottom of page 21, lines 6-7 of page 25 and lines 7-11 from the bottom of page 31.

Minor:

On page 7 TRPML1 is spelled TRMPM1 on the second sentence from the bottom of the page.

Ans) We are sorry for our mistake. The spelling was corrected.

Figure 4d shows a merge image at 4 hours post PA treatment however the only channel shown is green and there are no nuclei stained with dapi.

Ans) The pictures in Fig. 4d were changed with those containing DAPI staining, as suggested.

As Oregon Green 488 BAPTA-1 Dextran is sensitive to pH, the authors need to ensure that the changes of $[Ca^{2+}]_{Lys}$ with this technique is not due to changes in pH.

Ans) As suggested by the reviewer, we determined lysosomal pH after treatment with palmitic acid, Rot or O/A. PA treatment for 1 h did not induce significant changes of lysosomal pH. Rotenone treatment for 1 h also did not induce significant changes of lysosomal pH, while O/A treatment moderately increased lysosomal pH. Because $[Ca^{2+}]_{Lys}$ was increased by treatment with PA, rotenone or O/A for 1 h, we believe that changes of $[Ca^{2+}]_{Lys}$ is unrelated to changes of lysosomal pH. These results were incorporated in Supplementary Fig. 2a,b.

Reviewers' Comments:

Reviewer #1:

Remarks to the Author:

Thank you for your responses. I have no further comments on this manuscript

Reviewer #2:

Remarks to the Author:

The authors clearly showed that mitochondrial TFEB activation is followed by ROS production both in time course and from causal relationship by administration of the scavengers and, successfully excluded the possibility that PA could affect other signaling pathways except for TFEB, which fulfills our requirements.

In addition, the authors appropriately showed that the source of lysosomal Ca recovery is extracellular one, suggesting that Ca flux from extracellular milieu into lysosome through ER is important for TFEB-mediated mitophagy.

On the other hand, in the previous revise, we wondered if TFEB-mediated mitophagy in pancreatic beta cells is selective autophagy or due to promiscuous degradation of organelles by macroautophagy. The authors have shown that neither bulk macroautophagy nor ER-phagy is activated by mitochondrial stressors. These data are apparently supportive for the authors' claim, however, most papers cited there show that mitochondrial stressors such as HFD alter autophagic flux either in bulk or organ-specific manner in different contexts. For example, Hernández-Cáceres et al. reported that palmitic acid impairs autophagic flux in neuronal cells (Mol. Cell. Oncol. 25, 1789418, 2020), whereas Tong et al. demonstrated that autophagic flux is activated following HFD in heart (Cir Res 124:1360, 2019). From these points of view, we still need the data indicating that mitochondria in pancreatic-beta cells are selectively degraded in vivo by showing neither degradation of other organelles nor bulk macroautophagy is activated in TFEB-dependent manner, estimated by electron microscopy as shown in Sup. Fig 9a.

Furthermore, it is still unclear why TFEB regulates mitophagy in organelle-specific manner, though TFEB is considered as a master regulator of bulk macro-autophagy. The authors insisted that TFEB-mediated expression of the mitophagy receptors be involved in the regulation, which has not been fully proven yet. To support their idea, they should show that deletion of those receptors suppresses the induction of TFEB-mediated mitophagy.

Finally, the authors considered mitophagy described in this study as Parkin-dependent. However, previous studies showed that deficiency of Parkin/PINK1 axis in beta-cells does not exacerbate glucose tolerance (Open Biol. 7;4(5):140051, 2014. J Biol Chem. 3; 294(18): 7296–7307, 2019). If authors insist that TFEB-mediated mitophagy depends on Parkin/PINK-1 axis, authors should argue the mechanism in which deletion of Parkin/PINK-1 does not exacerbate beta cell function. Finally, the authors should re-consider the title of this manuscript, which is too concise and abstractive to describe its contents.

Reviewer #3:

Remarks to the Author:

The manuscript by Kihyoun Park et al. has been improved with the additional data and revisions, which has addressed most of my previous concerns. The manuscript determines how TFEB activation regulates beta-cell mitochondrial function.

Reviewer #1 (Remarks to the Author):

Thank you for your responses. I have no further comments on this manuscript

Ans) I would like to thank reviewer for positive response.

Reviewer #2 (Remarks to the Author):

The authors clearly showed that mitochondrial TFEB activation is followed by ROS production both in time course and from causal relationship by administration of the scavengers and, successfully excluded the possibility that PA could affect other signaling pathways except for TFEB, which fulfils our requirements.

In addition, the authors appropriately showed that the source of lysosomal Ca recovery is extracellular one, suggesting that Ca flux from extracellular milieu into lysosome through ER is important for TFEB-mediated mitophagy.

On the other hand, in the previous revise, we wondered if TFEB-mediated mitophagy in pancreatic beta cells is selective autophagy or due to promiscuous degradation of organelles by macroautophagy. The authors have shown that neither bulk macroautophagy nor ER-phagy is activated by mitochondrial stressors. These data are apparently supportive for the authors' claim, however, most papers cited there show that mitochondrial stressors such as HFD alter autophagic flux either in bulk or organ-specific manner in different contexts. For example, Hernández-Cáceres et al. reported that palmitic acid impairs autophagic flux in neuronal cells (Mol. Cell. Oncol. 25, 1789418, 2020), whereas Tong et al. demonstrated that autophagic flux is activated following HFD in heart (Cir Res 124:1360, 2019). From these points of view, we still need the data indicating that mitochondria in pancreatic-beta cells are

selectively degraded in vivo by showing neither degradation of other organelles nor bulk macroautophagy is activated in TFEB-dependent manner, estimated by electron microscopy as shown in Sup. Fig 9a.

Ans) We agree with the reviewer's suggestion that changes of autophagic flux after HFD feeding is variable depending on the duration of HFD feeding and tissue. Tong's paper showed increased mitophagic activity in cardiac tissue after HFD feeding, consistent with our results. They also showed that while bulk autophagic flux in cardiac tissue was increased at 6 weeks after HFD feeding but returned to basal level after 2 months of HFD feeding. In our study, we assessed mitophagic activity and bulk autophagic activity in pancreatic islets after 3 months of HFD feeding. And our results (increased mitophagic activity but no changes of bulk autophagic activity) are very similar to the Tong et al's paper, which might be related to prolonged duration of HFD. While tissues studied were different, these results suggest the autophagic activity could be different depending on the target organelles.

We expanded our previous EM studies to count the number of mitophagy and autophagy in 7~8 fields of magnified images (x 10,000) per mouse as recommended by the reviewer to further confirm that mitophagic activity is increased but bulk autophagic activity is not increased by HFD feeding for 3 months, which was incorporated as new Supplementary Fig. 9b and Supplementary Fig. 9d.

We also changed the sentence from "autophagic activity determined by colocalization between LAMP2 and LC3 was neither increased by HFD feeding nor decreased by *Tfeb* KO" to "autophagic activity determined by colocalization between LAMP2 and LC3 was not significantly changed by HFD feeding or *Tfeb* KO" (line 15~16 of page 14) to show our results more clearly.

Furthermore, it is still unclear why TFEB regulates mitophagy in organelle-specific manner,

though TFEB is considered as a master regulator of bulk macro-autophagy. The authors insisted that TFEB-mediated expression of the mitophagy receptors be involved in the regulation, which has not been fully proven yet. To support their idea, they should show that deletion of those receptors suppresses the induction of TFEB-mediated mitophagy.

Ans) As recommended by the reviewer, we conducted knockdown of *Ndp52* and *Optn*.

Knockdown of *Ndp52* or *Optn* significantly reduced mitophagic activity after treatment with palmitic acid or mitochondrial stressors (rotenone and oligomycin/antimycin A) determined by mito-Keima, indicating that increase of mitophagic activity by palmitic acid or mitochondrial stressors could be attributable to the induction of *Ndp52* and *Optn*. These results were incorporated as new Supplementary Fig. 11a~e.

Finally, the authors considered mitophagy described in this study as Parkin-dependent.

However, previous studies showed that deficiency of Parkin/PINK1 axis in beta-cells does not exacerbate glucose tolerance (Open Biol. 7;4(5):140051, 2014. J Biol Chem. 3; 294(18): 7296–7307, 2019). If authors insist that TFEB-mediated mitophagy depends on Parkin/PINK-1 axis, authors should argue the mechanism in which deletion of Parkin/PINK-1 does not exacerbate beta cell function.

Ans) We agree with the reviewer's comment that the role of PINK1/Parkin in β -cell function needs to be discussed in more detail. Thus, we expanded our discussion to include mild glucose intolerance observed after HFD feeding of β -cell *Parkin*-KO mice for a prolonged period of 16 weeks described in {Corsa CA et al. J Biol Chem. 3; 294(18): 7296–7307, 2019} and subtle mitochondrial functional changes observed in *PINK1*-KO β -cells {Deas E et al. Open Biol. 7;4(5):140051, 2014}. We also discussed findings in another paper showing the role of Parkin-mediated mitophagy in the maintenance of mitochondrial function and insulin release in streptozotocin-treated β -cells (Hoshino A et al. PNAS 111:3116, 2014). We also

discussed the types of Parkin-independent mitophagy in more detail that can occur in *Parkin*-KO β -cells as compensatory changes, and also previously reported defective mitochondrial function and insulin release from *Parkin*-knockdown INS-1 cells that could be due to the absence of such compensatory changes associated with *Parkin* KO (Kim HS et al. Mol Cell Endocrinol 382:174, 2014). These were incorporated in line 6 from the bottom of page 20 ~ line 2 of page 21.

Finally, the authors should re-consider the title of this manuscript, which is too concise and abstractive to describe its contents.

Ans) We agree with the comment that the title is abstractive and does not represent our results well. We changed the title to “Essential role of lysosomal Ca^{2+} -mediated TFEB activation in mitophagy and functional adaptation of pancreatic β -cells to metabolic stress”.

Reviewer #3 (Remarks to the Author):

The manuscript by Kihyoun Park et al. has been improved with the additional data and revisions, which has addressed most of my previous concerns. The manuscript determines how TFEB activation regulates beta-cell mitochondrial function.

Ans) I would like to thank reviewer for positive response.

Reviewers' Comments:

Reviewer #2:

Remarks to the Author:

The manuscript by Kihyoun Park et al. has been much improved with the additional data and we appreciate their effort to answer our questions. We think that they addressed most of our previous concerns. Regarding the issue about mitophagy, the data estimated by electron microscopy showed that mitochondria in pancreatic-beta cells are selectively degraded in vivo in TFEB-dependent manner. The authors also demonstrated that deletion of Ndp52 or Optn suppresses the induction of TFEB-mediated mitophagy, which has addressed our concerns in our previous comment. Thus, we have no further comments on this point. Regarding the request for title change, I have also no further comments.

On the other hand, even though the authors cited the several papers insisting that Pink1-Parkin is involved in mitophagy in pancreatic beta cells followed by glucose intolerance, these does not always support the authors' perspective that Pink1-Parkin system plays an important role in preserving homeostasis of pancreatic beta cells through mitophagy, especially in vivo (see the GTT data shown in reference #57). We propose that this part of the discussion should be modified by implying involvement of Pink1-Parkin independent mitophagy or deleting the inappropriate reference, or the authors showed the essential role of Pink1-Parkin system on mitophagy in beta cells by their hands.

Reviewer #2 (Remarks to the Author):

The manuscript by Kihyoun Park et al. has been much improved with the additional data and we appreciate their effort to answer our questions. We think that they addressed most of our previous concerns. Regarding the issue about mitophagy, the data estimated by electron microscopy showed that mitochondria in pancreatic-beta cells are selectively degraded in vivo in TFEB-dependent manner. The authors also demonstrated that deletion of Ndp52 or Optn suppresses the induction of TFEB-mediated mitophagy, which has addressed our concerns in our previous comment. Thus, we have no further comments on this point. Regarding the request for title change, I have also no further comments.

On the other hand, even though the authors cited the several papers insisting that Pink1-Parkin is involved in mitophagy in pancreatic beta cells followed by glucose intolerance, these does not always support the authors' perspective that Pink1-Parkin system plays an important role in preserving homeostasis of pancreatic beta cells through mitophagy, especially in vivo (see the GTT data shown in reference #57). We propose that this part of the discussion should be modified by implying involvement of Pink1-Parkin independent mitophagy or deleting the inappropriate reference, or the authors showed the essential role of Pink1-Parkin system on mitophagy in beta cells by their hands.

Ans) As suggested, we deleted unnecessary comments about Parkin and PINK-induced mitophagy (lines 15 ~ 21 of page 2 of previous submission). We previously commented about Parkin-independent mitophagy in line 10 ~ 12 of page 20. Again as suggested, we added PINK1-independent mitophagy (line 12 ~ 13 of page 20).